# TWICE SEQUENTIAL MONTE CARLO TREE SEARCH

## ABSTRACT

Model-based reinforcement learning (RL) methods that leverage search are responsible for many milestone breakthroughs in RL. Sequential Monte Carlo (SMC) recently emerged as an alternative to the Monte Carlo Tree Search (MCTS) algorithm which drove these breakthroughs. SMC is easier to parallelize and more suitable to GPU acceleration. However, it also suffers from large variance and path degeneracy which prevent it from scaling well with increased search depth, i.e., increased sequential compute. To address these problems, we introduce Twice Sequential Monte Carlo Tree Search (TSMCTS). Across discrete and continuous environments TSMCTS outperforms the SMC baseline as well as a popular modern version of MCTS. Through variance reduction and mitigation of path degeneracy, TSMCTS scales favorably with sequential compute while retaining the properties that make SMC natural to parallelize.

## 1 INTRODUCTION

The objective of Reinforcement Learning (RL) is to approximate optimal policies for decision problems formulated as interactive environments. For this purpose, model-based RL algorithms that use *search* (also called planning) with a model of the environment's dynamics for policy optimization have been tremendously successful. Examples include games (Silver et al., 2016), robotics (Hubert et al., 2021) and algorithm discovery (Fawzi et al., 2022; Mankowitz et al., 2023). These milestone approaches are all based in the Alpha/MuZero (A/MZ, Silver et al., 2018; Schrittwieser et al., 2020) algorithm family and are driven by Monte Carlo Tree Search (MCTS, see Świechowski et al., 2023).

Like many search algorithms, the main bottleneck of MCTS is intensive compute and therefore runtime cost. Due to the sequential nature of MCTS (Liu et al., 2020; Macfarlane et al., 2024), it is challenging to address its runtime cost through parallelization and GPU acceleration (for example, with JAX, Bradbury et al., 2018) which are staples of other modern deep RL approaches. In addition, MCTS requires maintaining the entire search tree in memory. Modern GPU-acceleration approaches such as JAX require static shapes for best performance which forces memory usage to scale with the tree size and makes space complexity another possible bottleneck for GPU scalability.

To address this, alternative search algorithms have emerged (Piché et al., 2019). These algorithms use Sequential Monte Carlo (SMC, see Chopin & Papaspiliopoulos, 2020) for policy optimization in the Control as Inference (CAI, see Levine, 2018) probabilistic inference framework for RL. SMC is used to approximate a distribution over trajectories generated by an improved policy at the root using $N$ particles in parallel. The parallel nature and lower memory cost, which scales linearly with $N$, make SMC well suited for parallelization and GPU acceleration, as demonstrated by Macfarlane et al. (2024), which has also shown that SMC is competitive with MCTS for policy improvement.

SMC however suffers from two major problems: sharply increasing variance with search depth and path degeneracy (Chopin & Papaspiliopoulos, 2020). The variance increase stems from the exponential growth in the number of possible trajectories $s_{1:T}$ in the search depth $T$. Path degeneracy is a phenomenon where due to resampling eventually all particles become associated with the same state-action at the root of the search tree. This renders any additional search completely obsolete and collapses the root policy into a delta distribution causing target degeneracy (de Vries et al., 2025). These problems can cause the performance of SMC to *deteriorate* rather than *scale* with sequential compute (search depth). In contrast, MCTS scales well with sequential compute and does not suffer from path degeneracy.

To address these limitations of SMC we design a novel search algorithm which we call Twice Sequential Monte Carlo Tree Search (TSMCTS). We begin with a reformulation of SMC for RL which generalizes beyond the framework of CAI, simplifying the analysis and surfacing connections to MCTS. To mitigate policy target variance and degeneracy we switch the perspective of the search from estimating trajectories to estimating the value of an improved policy at the root. This facilitates incorporating the backpropagation mechanism of MCTS for value aggregation at the root. We call this intermediate algorithm SMC Tree Search (SMCTS). Building on SMCTS, TSMCTS utilizes Sequential Halving (Karnin et al., 2013) for better search resource allocation at the root. The resulting algorithm sequentially calls SMCTS at the root on a halving number of actions with doubling number of particles, in parallel (thus, twice sequential). This addresses the remaining effects of path degeneracy at the root while acting as an additional variance reduction mechanism.

We evaluate TSMCTS on a range of continuous and discrete environments, where it significantly outperforms the SMC baseline as well as a popular modern version of MCTS (GumbelMCTS, Danihelka et al., 2022). TSMCTS scales well with additional sequential compute, unlike the SMC baseline which deteriorates, while maintaining the same space and runtime complexity properties that make SMC well suited for parallelization. In ablations, we verify empirically that TSMCTS demonstrates significantly reduced estimator variance and mitigates path degeneracy.

## 2 BACKGROUND

In RL, the environment is represented by a Markov Decision Process (MDP, Bellman, 1957) $\mathcal{M} = \langle \mathcal{S}, \mathcal{A}, \rho, R, P, \gamma \rangle$. $\mathcal{S}$ is a set of states, $\mathcal{A}$ a set of actions, $\rho$ an initial state distribution, $R : \mathcal{S} \times \mathcal{A} \to \mathbb{R}$ a bounded possibly stochastic reward function, and $P$ is a transition distribution such that $P(s'|s, a)$ specifies the probability of transitioning from state $s$ to state $s'$ with action $a$. The policy of the agent $\pi \in \Pi$ is defined as a distribution over actions $a \sim \pi(s)$ and its optimality is defined with respect to the objective $J_\pi$, the maximization of the *expected discounted return* (also called value $V^\pi$):

$$J_\pi = \mathbb{E}[V^\pi(s_0)|s_0 \sim \rho] = \mathbb{E}\Big[\sum_{t=0}^{H-1} \gamma^t R(s_t, a_t)\Big|s_0 \sim \rho, s_{t+1} \sim P(s_t, a_t), a_t \sim \pi(s_t)\Big]. \quad (1)$$

The discount factor $0 < \gamma < 1$ is used in infinite-horizon MDPs, i.e. $H = \infty$, to guarantee that the values remain bounded. A state-action *Q-value function* is defined as follows: $Q^\pi(s, a) = \mathbb{E}[R(s, a) + \gamma V^\pi(s')| s' \sim P(s, a)]$. We denote the value of the optimal policy $\pi^*$ with $V^*(s) = \max_\pi V^\pi(s), \forall s \in \mathcal{S}$. In model-based RL (MBRL) the agent uses a model of the dynamics of the environment $(P, R)$ to optimize its policy, often using search algorithms such as MCTS or SMC.

**Policy improvement** is used to motivate the convergence of approximate policy iteration algorithms to the optimal policy (see Danihelka et al., 2022; Oren et al., 2025b). We will prove that our formulation of SMC for RL approximates policy improvement and can be used in a similar manner to MCTS. We define policy improvement operators $\mathcal{I} : \Pi \times \mathcal{Q} \to \Pi$ as any operator such that $\forall s \in \mathcal{S} : V^{\mathcal{I}(\pi, Q^\pi)}(s) \geq V^\pi(s)$ and $\exists s \in \mathcal{S} : V^{\mathcal{I}(\pi, Q^\pi)}(s) > V^\pi(s)$, unless $\pi$ is already an optimal policy. We define $\mathcal{Q}$ generally as the set of all bounded functions on the state-action space $q \in \mathcal{Q} : \mathcal{S} \times \mathcal{A} \to \mathbb{R}$, to indicate that policy improvement operators are defined for approximate $q \approx Q^\pi$ and exact $Q^\pi$.

**Greedification** The *policy improvement theorem* (Sutton & Barto, 2018) proves that *greedification* (Chan et al., 2022; Oren et al., 2025b) produces policy improvement when applied with respect to a policy $\pi$ and its value $Q^\pi$. Greedification operators $\mathcal{I}$ are operators over the same space, such that the policy $\mathcal{I}(\pi, q)(a|s)q(s, a)$ is *greedier* than $\pi$ with respect to $q$, defined as the follows:

$$\forall s \in \mathcal{S} : \quad \sum_{a \in A} \mathcal{I}(\pi, q)(a|s)q(s, a) \geq \sum_{a \in A} \pi(a|s)q(s, a), \quad (2)$$

$$\exists s \in \mathcal{S} : \quad \sum_{a \in A} \mathcal{I}(\pi, q)(a|s)q(s, a) > \sum_{a \in A} \pi(a|s)q(s, a), \quad (3)$$

unless $\pi$ is already a greedy ($\arg\max$) policy with respect to $q$. We define *strict* greedification operators $\mathcal{I}$ as operators that satisfy a strict $>$ inequality 2, unless $\pi$ is already a greedy policy at $s$. A

popular strict greedification operator is that of *regularized policy improvement* (Grill et al., 2020):

$$\mathcal{I}_{GMZ}(\pi, q)(a|s) = \frac{\exp(\beta q(s, a) + \log \pi(a|s))}{\sum_{a' \in \mathcal{A}} \exp(\beta q(s, a') + \log \pi(a'|s))} \propto \pi(a|s) \exp\left(\beta q(s, a)\right) \quad (4)$$

which trades off with an inverse-temperature parameter $\beta$ between *greedification* (maximizing $\sum_{a \in \mathcal{A}} \pi(a|s)q(s, a)$ with respect to $\pi$) and regularization with respect to the prior policy $\pi$. We will use greedification operators to drive the policy improvement produced by SMC and TSMCTS.

**Monte Carlo Tree Search** (MCTS) is used in RL to select actions in the environment and to produce targets for training in the form of policy improvement and value bootstraps. MCTS uses a model of the environment (either exact, as in AlphaZero (AZ), or learned, latent and/or approximate, as in MuZero (MZ)) to construct a search tree where each node is associated with a state $s \in \mathcal{S}$. The root is set to the current state of the environment $s_0 := s$. For convenience, we will use the subscript $s_t$ to denote states in the planner (here MCTS and later SMC), and will clarify when not clear from context whether it refers to states in the environment or the planner. Each node $s_t$ maintains: (i) a prior-policy $\pi_\theta(s_t)$. (ii) The mean reward $r(s_t, a)$ for each visited action $a$. (iii) An estimate of the value $V_M(s_t)$ which is computed as the average of all $M$ returns passed through this node.

MCTS repeats a three-step process: *search*, *expansion* and *backpropagation*. The tree is traversed following a *search policy* $\pi'$ until a non-expanded node $s_t$ is reached. Inspired by the work of Grill et al. (2020), modern algorithms such as GumbelMuZero (GMZ, Danihelka et al., 2022) use $\pi' = \mathcal{I}_{GMZ}(\pi_\theta, Q_M)$ (Equation 4) with a $\beta$ parameter that increases with $M$, the number of visitations to the node. Once a non-expanded node $s_t$ has been reached, the node is *expanded* by sampling an action $a_t$ from the prior policy $\pi_\theta$, expanding the transition $r_t = \mathbb{E}[R(s_t, a_t)], s_{t+1} \sim P(s_{t+1}|s_t, a_t)$ (which is traditionally deterministic) and evaluating $Q^{\pi_\theta}(s_t, a_t) \approx r_t + \gamma V^{\pi_\theta}(s_{t+1})$. $V^{\pi_\theta}(s_{t+1})$ is usually approximated with a value DNN $v_\phi \approx V^{\pi_\theta}$. The new evaluation is then *backpropagated* up the search tree, through all nodes along the trajectory $\tau_t = s_0, a_0, \ldots, s_t, a_t, s_{t+1}$, updating the running average of the value estimates: $V_{M+1} = \frac{1}{M+1} \sum_{i=1}^{M+1} \nu_i$, where $\nu_i = \sum_{j=0}^{t} \gamma^j r_j^i + \gamma^{t+1} v_\phi(s_{t+1}^i)$. This process is repeated $B$ times, the search budget of the algorithm.

When MCTS terminates, an action is selected at the root using an improved policy $\pi_{improved}$. To drive an approximate policy iteration loop, Danihelka et al. (2022) use $\pi_{improved} := \mathcal{I}_{GMZ}(\pi_\theta, Q_M)(s_0)$, where $Q_M(s_0, a) = r(s_0, a) + \gamma \mathbb{E}_{P(s_1|s_0, a)}[V_M(s_1)]$. $\pi_{improved}(s_0)$ is used to train the prior policy $\pi_\theta$ using a cross-entropy loss. The value at the root $V_M(s_0)$ is used to produce bootstraps for TD-targets (Schrittwieser et al., 2020) or value targets directly (Oren et al., 2025a).

**Sequential Halving with MCTS**   Due to the compute budget $B$ being known in advance in many cases in practice, in GumbelAlpha/MuZero (GA/MZ, Danihelka et al., 2022), the authors propose to separate MCTS to two processes: a *simple-regret* minimization at the root $s_0$ through the Sequential-Halving (SH, Karnin et al., 2013) algorithm. At all other nodes the original MCTS process is used. SH begins with a set $|A_1| = m_1$ of actions to search and a total search budget $B$. SH then divides the search budget equally across $i = 1, \ldots, \log_2 m_1$ iterations. The per-iteration budget itself is divided equally across the actions searched this iteration $A_i$. As its name suggests, SH halves the number of actions that are searched each iteration by taking the top half according to a certain statistic, $\arg \operatorname{top} \mathcal{I}_{GMZ}(\pi_\theta, Q_M)(s_0)$ in the case of GA/MZ. As a result, at each iteration the search budget for the remaining actions doubles. After the final iteration the algorithm returns the improved policy $\pi_{improved}(s_0) = \mathcal{I}_{GMZ}(\pi_\theta, Q_{\log_2 m_1})(s_0)$, and the value of the root state $V_{search}(s_0) = \sum_{a \in A_1} \pi_{improved}(a|s_0)Q_{\log_2 m_1}(s_0, a)$.

**Sequential Monte Carlo** (SMC) methods approximate a sequence of *target distributions* $p_t(x_{0:t})$ using *proposal distributions* $u_t(x_t \mid x_{0:t-1})$. At each time step $t \in \{0, \ldots, T\}$, $N$ particles $x_t^n$ with weights $w_t^n$ are updated via *mutation, correction, and selection* (Chopin, 2004). *Mutation*: each trajectory $x_{0:t-1}^n$ is extended by sampling $x_t^n \sim u_t(x_t \mid x_{0:t-1}^n)$. *Correction*: The weights are updated to account for the target distribution, such that the set of weighted particles $\{x_t^n, w_t^n\}_{n=1}^N$ approximates expectations under the target:

$$w_t^n = w_{t-1}^n \cdot \frac{p_t(x_t^n \mid x_{0:t-1}^n)}{u_t(x_t^n \mid x_{0:t-1}^n)}, \qquad \frac{\sum_{n=1}^N w_t^n f(x_t^n)}{\sum_{n=1}^N w_t^n} \approx \mathbb{E}_{p_t}[f(x_t)], \quad (5)$$

where $f(x_t)$ is any function of interest. *Selection*: The particles are resampled proportionally to the normalized weights: $\{x_t\}_{n=1}^N \sim \text{Multinomial}(N, \text{normalized } w_t), \{w_t^n = 1\}_{n=1}^N$ to prevent particle degeneracy. We refer to Chopin & Papaspiliopoulos (2020) for more details.

**SMC as a search algorithm for RL**    Piché et al. (2019) use SMC as a search algorithm by defining the target distribution $p_t(\tau_t)$ over trajectories $\tau_t = (s_0, a_0, \ldots, s_t, a_t, s_{t+1}) = x_{0:t}$ (superscripted $\tau_t^n$ to denote trajectory per particle). The target is conditioned on an optimality variable $\mathcal{O}_{1:H}$, such that $p(\mathcal{O}_{1:H} \mid \tau_H) \propto \exp\left(\sum_{t=1}^H r_t\right)$, following the control-as-inference (CAI) framework (see Levine, 2018), up to a horizon $H$. The proposal distribution is defined using a prior policy $\pi_\theta$, while the target distribution incorporates the soft-optimal policy $\mu$ and the soft-value function $V_{\text{soft}}$:

$$u_t(\tau_t \mid \tau_{t-1}) = P(s_t \mid s_{t-1}, a_{t-1})\,\pi_\theta(a_t \mid s_t), \tag{6}$$

$$p_t(\tau_t \mid \tau_{t-1}) \propto P(s_t \mid s_{t-1}, a_{t-1})\,\mu(a_t \mid s_t)\,\mathbb{E}_{s_{t+1}\mid s_t, a_t}\big[\exp(A_{\text{soft}}(s_t, a_t, s_{t+1}))\big], \tag{7}$$

$$w_t^n = w_{t-1}^n \frac{p_t(\tau_t^n \mid \tau_{t-1}^n)}{u_t(\tau_t^n \mid \tau_{t-1}^n)} \propto w_{t-1}^n \frac{\mu(a_t^n \mid s_t^n)}{\pi_\theta(a_t^n \mid s_t^n)}\,\mathbb{E}_{s_{t+1}^n\mid s_t^n, a_t^n}\big[\exp(A_{\text{soft}}(s_t^n, a_t^n, s_{t+1}^n))\big], \tag{8}$$

where $A_{\text{soft}}(s_t, a_t, s_{t+1}) = r_t + V_{\text{soft}}(s_{t+1}) - \log \mathbb{E}_{s_t \mid s_{t-1}, a_{t-1}} V_{\text{soft}}(s_t)$. See (Piché et al., 2019) for derivation. We refer to this algorithm as CAI-SMC to distinguish from other variations. In the maximum entropy setup, $\mu$ is a uniform policy, which recovers the maximum entropy solution (Haarnoja et al., 2018). $V_{\text{soft}}$ is learned using a deep neural network trained with a temporal-difference loss. Piché et al. (2019) train the policy $\pi_\theta$ using Soft Actor Critic (Haarnoja et al., 2018). The policy returned by CAI-SMC is only used to select actions in the environment. The model used by the planner is learned from interactions.

Macfarlane et al. (2024) showed that CAI-SMC can be used as a policy improvement operator in a manner similar to that in which MCTS is used by AZ, in their method SPO. SPO uses the SMC planner derived by Piché et al. (2019) (CAI-SMC) with $\mu = \pi_\theta$ which facilitates an Expectation-Maximization framework and allows the policy to concentrate over time to the true optimal policy, rather than the soft-optimal policy of CAI.

## 3    Sequential Monte Carlo Search for Reinforcement Learning

We begin by extending Piché et al. (2019)'s formulation of SMC as a search algorithm for RL beyond the framework of CAI. This formulation is simpler, accepts general improvement operators $\mathcal{I}$ and facilitates a perspective shift from *reasoning over a distribution over trajectories* to *reasoning over the values of actions from a mixture of improved policies at the root* which we will build on in the following sections. Similar to Piché et al. (2019), we formulate the proposal $u_t(\tau_t)$ and target $p_t(\tau_t)$ distributions as distributions over trajectories $\tau_t = s_0, a_0, \ldots, s_t, a_t, s_{t+1}$. We define the proposal distribution $u_t(\tau_t)$ as the distribution induced by some prior policy $\pi_\theta$:

$$u_t(\tau_t) = \rho(s_0)\Pi_{i=0}^t P(s_{i+1}\mid s_i, a_i)\pi_\theta(a_i\mid s_i) \;\Rightarrow\; u_t(\tau_t\mid \tau_{t-1}) = P(s_{t+1}\mid s_t, a_t)\pi_\theta(a_t\mid s_t). \tag{9}$$

We define the target distribution $p_t(\tau_t)$ as the distribution induced by an improved policy $\pi' = \mathcal{I}(\pi_\theta, Q^\pi)$ for some policy improvement operator $\mathcal{I}$:

$$p_t(\tau_t) = \rho(s_0)\Pi_{i=0}^t P(s_{i+1}\mid s_i, a_i)\pi'(a_i\mid s_i) \;\Rightarrow\; p_t(\tau_t\mid \tau_{t-1}) = P(s_{t+1}\mid s_t, a_t)\pi'(a_t\mid s_t). \tag{10}$$

Given $p_t(\tau_t)$ and $u_t(\tau_t)$, the importance sampling weights $w_t^n$ for SMC derive as follows:

$$w_t^n = w_{t-1}^n \frac{p_t(\tau_t^n\mid \tau_{t-1}^n)}{u_t(\tau_t^n\mid \tau_{t-1}^n)} = w_{t-1}^n \frac{P(s_{t+1}^n\mid s_t^n, a_t^n)\pi'(a_t^n\mid s_t^n)}{P(s_{t+1}^n\mid s_t^n, a_t^n)\pi_\theta(a_t^n\mid s_t^n)} = w_{t-1}^n \frac{\pi'(a_t^n\mid s_t^n)}{\pi_\theta(a_t^n\mid s_t^n)} \tag{11}$$

In practice, the value $Q^\pi(s, a)$ used to compute the improved policy $\pi'$ is approximated with DNNs $q_\phi(s, a)$ or $r(s, a) + \gamma v_\phi(s')$ like in CAI-SMC and A/MZ. We refer to this formulation as **RL-SMC** (Algorithm 2). Equation 11 reduces to Equation 8 for the soft-advantage operator of CAI-SMC (see Appendix A.2 for full derivation).

**Policy improvement at the root**    Like CAI-SMC, RL-SMC produces a policy $\hat{\pi}_{SMC}^T$ at the root $s_0$ after $T$ steps with empirical occupancy counts using the particles:

$$\hat{\pi}_{SMC}^T(a\mid s_0) := \frac{1}{N}\sum_{n=1}^N \mathbb{1}_{\tau_T^n(a_0)=a} \approx \mathbb{P}\big(\tau_T(a_0) = a\big) =: \pi_{SMC}^T(a\mid s_0), \tag{12}$$

where $\tau_T(a_0)$ denotes the first action in the trajectory. We verify that RL-SMC approximates policy improvement so that it can drive an approximate policy iteration loop in a similar manner to MCTS:

**Theorem 1.** *For any improvement operator $\mathcal{I}$, search horizon $T$, prior policy $\pi_\theta$, true dynamics model $(P, R)$ and true evaluation $Q^{\pi_\theta}$ RL-SMC with infinite particles is a policy improvement operator.*

**Intuition** RL-SMC produces a distribution over trajectories $p_T(\tau_T)$ from a policy that is improved with respect to the prior policy $\pi_\theta$ at states $\{s_0, \ldots, s_T\}$. Since this policy is improved with respect to the future $\{s_0, \ldots, s_{T+1}\}$, it is of course also improved at $s_0$, the current state in the environment. See Appendix A.1 for a complete proof.

The proof of Theorem 1 points to one of the advantages of using search for policy improvement compared to model-free approaches. By unrolling with the model, RL-SMC produces a policy that is improved for $T$ *consecutive time steps*, in contrast to the single step of model free methods:

**Corollary 1.** *For any strict improvement operator $\mathcal{I}$, search horizon $T$, prior policy $\pi_\theta$, true dynamics model $(P, R)$ and true evaluation $Q^{\pi_\theta}$ the policy produced by RL-SMC satisfies:*

$$V^{\pi_{SMC}^T}(s_0) > V^{\pi_{SMC}^{T-1}}(s_0) > \cdots > V^{\pi_{SMC}^1}(s_0) > V^{\pi_\theta}(s_0) \tag{13}$$

*as long as $\pi_\theta$ is not already an $\arg\max$ policy with respect to $Q^{\pi_\theta}$ at all states $s_0, \ldots, s_T$.*

The proof follows directly from applying strict improvement operators (improvement operators that satisfy a strict $>$ inequality 2 at all states unless the policy is already an $\arg\max$ policy).

However, the root estimator $\hat{\pi}_{SMC}^T(s_0)$ suffers from two major problems: variance that grows sharply in $T$ and path degeneracy (see Chopin, 2004; Chopin & Papaspiliopoulos, 2020).

**Large variance** The variance of SMC can scale up to polynomially with depth $t$, in order $\mathcal{O}(t^\Omega)$, where $\Omega$ is the dimension of the domain of the target distribution, $p_t(\tau_t)$ (Chopin, 2004). In RL/CAI-SMC however the dimension of the domain $\tau_t$ itself grows linearly with $t$: $\Omega_t = d_{s,a}t$, where $d_{s,a}$ is the joint dimension of the state-action space $\mathcal{S}, \mathcal{A}$ (for example if $s \in \mathbb{R}^5, a \in \mathbb{R}^2$ then $d_{s,a} = 7$). As a result, the variance of the estimator can increase up to *super-exponentially* in $t$: $\mathcal{O}(t^{td_{s,a}})$.

**Path degeneracy** Consecutive selection steps $t$ are likely to concentrate all particles $i$ to trajectories that are associated with one root action $a_0^i$. Once all particles are associated with the same root action $a_0^i$, say at a step $h$, the estimator $\hat{\pi}_{SMC}^t(a_0^i|s_0) = 1$ and zero for all other root actions $a_0 \neq a_0^i$. From that point on, the estimator will not change for all depth $t > h$. This is problematic for two reasons: (i) The search has no effect from $t > h$, and the algorithm cannot scale with additional sequential compute (increasing $T$). This is because particles will not be resampled out of trajectories starting in action $a_0^i$ and therefore, $\hat{\pi}_{SMC}^t(a_0^i|s_0)$ will not change for $t > h$. (ii) It results in a delta distribution policy target at the root $s_0$ that is a crude approximation for any underlying improved policy $\pi_{improved}(s_0)$ but an $\arg\max$.

Unlike RL-SMC, MCTS treats the search problem as the problem of identifying the best action at the root using value estimates $Q_M(s_0, \cdot) \approx Q^{\pi_{improved}}(s_0, \cdot)$, rather than a distribution over trajectories $p_t(\tau_t)$. By averaging the returns of all trajectories observed during search MCTS reduces the variance of the root estimator $Q_M$. Additionally, by maintaining a value estimate for each visited action at the root MCTS prevents the effects of path degeneracy: $Q_M$ updates with each search step, and the policy cannot collapse to a delta distribution, resulting in richer policy targets. This observation motivates the next step in the design of the algorithm: a value-based perspective on RL-SMC's search.

## 4 VALUE-BASED SEQUENTIAL MONTE CARLO

Maintaining estimates $Q^{\pi_{SMC}^t}(s_0, a)$ in addition to a distribution over trajectories from the root can address both of the problems caused by path degeneracy as discussed earlier: (i) The estimate $Q^{\pi_{SMC}^t}$ does not stop updating when all particles are associated with one action at $t = h$ and thus search for $t > h$ is not obsolete, allowing SMC to benefit from increased search depth. (ii) Information is not lost about actions that have no remaining particles, and thus, target degeneracy is prevented. This is similar to the idea recently proposed by de Vries et al. (2025), albeit in the guise of policy

log-probabilities in the framework of CAI. The value at the root $Q^{\pi^t_{SMC}}(s_0, \cdot)$ can be approximated using the particles:

$$Q^{\pi^t_{SMC}}(s_0, a_0) = \mathbb{E}_{\pi^t_{SMC}}\Big[\sum_{i=0}^{t} \gamma^i r_i + \gamma^{t+1} V^{\pi_\theta}(s_{t+1}) \,\big|\, s_0, a_0\Big] \tag{14}$$

$$\approx \sum_{n=1}^{N} w_t^n \mathbb{1}_{a_0^n = a_0} \sum_{i=0}^{t} \gamma^i r_i^n + \gamma^{t+1} V^{\pi_\theta}(s_{t+1}^n) := Q_t(s_0, a_0) \tag{15}$$

The estimator $Q_t(s_0, a_0)$ by itself however is potentially just as high variance as $\pi^t_{SMC}$. Instead, we can keep track of the *average* return observed during search, with a backpropagation step similar to MCTS: $\bar{Q}_t(s_0, a_0) = \frac{1}{t} \sum_{i=1}^{t} Q_i(s_0, a_0)$. Whenever there are no particles associated with action $a_0$, the value $\bar{Q}_t(s_0, a_0)$ is not updated. By mixing predictions for different steps $Q_1, \ldots, Q_t$, any errors that can average out now average out (see Appendix A.4 for more detail). On the other hand, although $Q_t$ is an unbiased estimate of $Q^{\pi^t_{SMC}}$, $\bar{Q}_t$ is not. Instead, $\bar{Q}_t$ estimates the value of a mixture of more and more improved policies $\pi^1_{SMC}, \ldots, \pi^t_{SMC}$. Since every policy $\pi^i_{SMC}$ in the mixture is already an improved policy, this is not a problem, it merely results in a value estimate of a less-improved (but still improved) policy than $\pi^T_{SMC}$.

This value-based extension to RL-SMC can be thought of as iterating: (i) *Search*: compute importance sampling weights to align with the improved policy $\pi'(s_t)$. (ii) *Backpropagation*: evaluate the returns for each particle at states $s_{t+1}$, average the return across all particles associated with the same action $a_0$ at the root and incorporate it into the running mean $\bar{Q}_t$. (iii) *Expansion*: sample from the prior-policy $\pi_\theta(s_{t+1})$. Due to the similarity between this three-step process and MCTS', we refer to this algorithm as Sequential-Monte-Carlo Tree Search (**SMCTS**, summarized in Algorithm 3).

**Policy improvement at the root**  To extract policy improvement at the root $\pi_{improved}(s_0)$ using the value estimates $\bar{Q}_T(s_0, \cdot)$, any policy improvement operator $\mathcal{I}$ can be chosen. SMCTS returns:

$$\pi_{improved}(s_0) = \mathcal{I}(\pi_\theta, \bar{Q})(s_0), \quad V_{search}(s_0) = \sum_{a \in A_0} \bar{Q}_T(s_0, a) \pi_{improved}(a|s_0). \tag{16}$$

One effect of path degeneracy remains however: all particles can still collapse to search only one root ancestor. In addition, SMCTS does not fully leverage the insight that the search objective is policy improvement *specifically* at the root. We address these next.

## 5 TWICE-SEQUENTIAL MONTE CARLO TREE SEARCH

One of the key observations of Danihelka et al. (2022) is that at the root of the search tree $s_0$, the search budget of the algorithm is known in advance. This motivates using known-budget-optimization algorithms such as SH (see Section 2) at the root of the search tree. By combining SH (Karnin et al., 2013) with SMCTS, we are able to further reduce estimator variance and mitigate remaining effects of path degeneracy at the root.

At each SH iteration $i$, SH resets the search back to the root. This results in repeated re-searching of actions at the root. By aggregating the value predictions $\bar{Q}_T^i$ of SMCTS across iterations $i$, SH induces further lower variance estimates of the value at the root. This is similar to existing methods to addressing variance in SMC such as *Ancestor Sampling* (Lindsten et al., 2014). Further, at each iteration $i$ SH reduces the number of searched actions while increasing the search budget per action. As a result, SH minimizes the variance of the estimator for the value-maximizing actions: the actions that are the most important for action selection and policy improvement. Finally, SH searches each action at the root independently in parallel, which mitigates the remaining effect of path degeneracy at the root. We formulate this Sequential-Halving Sequential-Monte-Carlo Tree Search algorithm, or *Twice Sequential Monte Carlo Tree Search* (**TSMCTS**), below.

**TSMCTS** requires a number of particles $N$, depth budget $T$, and a number of starting actions to search at the root $m_1$. The total search budget (number of model expansions) $B = NT$ is then the particle budget multiplied by the depth budget. The total number of iterations of SH is $\log_2 m_1$. SH assigns a compute budget $B_i$ per action at the root per iteration $i = 1, \ldots, \log_2 m_1$. $B_i$ can

be computed as follows: $B_i = \frac{NT}{m_i \log_2 m_1}$, where $m_{i+1} = m_i/2$, $i \geq 1$. In order to preserve the parallelizability properties of SMC we assign $N/m_i$ particles per-action per-iteration (we assume for simplicity that $m_i$ divides $N$ and otherwise round for a total particle budget of $N$ at each iteration). This results in the number of particles per-action per-iteration doubling every iteration: $N_{i+1} = 2N_i$. To maintain the same total compute cost $B = NT$ as SMC, at each iteration $i$ SH searches up to depth $T_{SH} < T$:

$$T_{SH} = \frac{B_i}{N/m_i} = \frac{NT}{m_i \log_2 m_1} \frac{m_i}{N} = \frac{T}{\log_2 m_1} < T. \tag{17}$$

Instead of searching to the full depth $T$, TSMCTS searches repeatedly to a lesser depth $T_{SH}$, and thus each individual estimator $\bar{Q}^i_{T_{SH}}$ is a lower variance estimator. This results in additional reduction in estimator variance in $T$, traded off against reduction in the search horizon which becomes $T_{SH}$.

At the first iteration $i = 1$, the set $A_1$ of $m_1$ actions to search are chosen as the top $m_1$ actions according to probabilities $\pi_\theta(s_0)$. To approximate sampling without replacement from the policy, in discrete action spaces we use the Gumbel-top-k trick (Kool et al., 2019), which adds noise from the Gumbel distribution $(g \in \mathbb{R}^{|\mathcal{A}|}) \sim \text{Gumbel}(0)$, $\pi(s_0) \propto \exp(\log \pi_\theta(s_0) + g)$.

At each iteration $i \geq 1$ TSMCTS executes SMCTS as a subroutine independently in parallel for each $a \in A_i$, the top $m_i$ ($i > 1: m_i = \frac{m_{i-1}}{2}$) actions at the root according to the current improved policy:

$$i = 1: A_1 = \arg \text{top}(\pi(s_0), m_1), \quad i > 1: A_i = \arg \text{top}(\mathcal{I}(\pi, Q^{i-1}_{SH})(s_0), m_i). \tag{18}$$

SMCTS returns the value of the improved policy at the next state for this iteration, $V^i_{SMCTS}(s_1)$. The value for each action at the root $a \in A_i$ is computed: $Q^i_{SMCTS}(s_0, a) = r(s_0, a) + \gamma V^i_{SMCTS}(s_1)$. As noted above, because the search budget per action doubles each iteration, $Q^i_{SMCTS}$ is a lower-variance estimator than $Q^{i-1}_{SMCTS}$ for all actions visited this iteration. To account for that we extend the computation of the value average across iterations $i$ to a weighted average. The average is weighted by the "visitations" - the number of particles - to this action this iteration:

$$\forall a \in A_i: \quad Q^i_{SH}(s_0, a) = \frac{1}{\sum_{j=1}^i N_j(a)} \sum_{j=1}^i N_j(a) Q^j_{SMCTS}(s_0, a), \tag{19}$$

where $N_i(a) \geq 0$ is the number of particles assigned to root action $a$ at iteration $i$ and $Q^i_{SMCTS}(s_0, a) := 0$ for root actions $a$ that were not searched at iteration $i$ (the term $Q^i_{SMCTS}(s_0, a)$ will be multiplied by $N_i(a) = 0$ for these actions and thus the actual value does not matter). In practice, we maintain two vectors of size $m_1$ of running sums:

$$N^i(a) := \sum_{j=1}^i N_j(a), \quad Q^i_{sum}(s_0, a) = \sum_{j=1}^i N_j(a) Q^j_{SMCTS}(s_0, a). \tag{20}$$

TSMCTS returns: (i) The improved policy at the root computed using the last iteration's Q-value: $\pi_{improved} = \mathcal{I}(\pi, Q^{\log_2 m_1}_{SH})$. (ii) An estimate of the value of the policy $V_{search}(s_0) = \sum_{a \in A_1} \pi_{improved}(a|s_0) Q^{\log_2 m_1}_{SH}(s_0, a)$. These outputs are used to train the value and policy networks in the same manner as SPO and A/MZ. That is, the improved policy $\pi_{improved}$ is used to train the policy $\pi_\theta$ using cross-entropy loss. The value estimate $V_{search}(s_0)$ is used to bootstrap value targets to train the critic $v_\phi$, as in (de Vries et al., 2025). Action selection is done by sampling from the improved policy during learning $a \sim \pi_{improved}(s_0)$ and deterministically taking the $\arg \max$ action during evaluation $a = \arg \max_{b \in A_1} \pi_{improved}(b|s_0)$. We refer to Appendix B for more details.

A more detailed derivation of Equation 19 and discussion of the variance reduction mechanisms are provided in Appendices A.3 and A.4 respectively. TSMCTS maintains the same space and runtime complexity of the RL-SMC baseline (see Appendix A.5). We summarize TSMCTS in Algorithm 4.

**Choice of operator** The operator $\mathcal{I}_{GMZ}$ was used by Danihelka et al. (2022) for search and policy improvement at the root in MCTS. $\mathcal{I}_{GMZ}$ intentionally balances between maximizing with respect to $Q$ while minimizing the divergence from $\pi_\theta$, making it a natural choice for TSMCTS as well.

## 6 RELATED WORK

SMC has been used in RL and more generally MDP solving for a variety of purposes (see (Lazaric et al., 2007; Hoffman et al., 2007; Le et al., 2018) for examples). Our focus in this section is on related work in the area of SMC for search in RL. Multiple works build upon Piché et al. (2019)'s derivation of CAI-SMC for search. Lioutas et al. (2023) extend the proposal distribution with a $Q$ critic, to direct the mutation step towards more promising trajectories. Macfarlane et al. (2024)'s approach of using CAI-SMC for policy improvement and benefiting from SMC's capacity to parallelize effectively across particles. de Vries et al. (2025) extends the SMC search further with trust-region optimization methods and additionally address terminal states with *revived resampling*. These advancements are orthogonal and natural to incorporate into RL-SMC and TSMCTS (see Figure 1 in Section 7). de Vries et al. (2025) also propose to address path degeneracy by essentially maintaining the *last* return observed for each action at the root, thus preventing the collapse of the improved policy at the root to a delta distribution. In contrast, SMCTS aggregates *all* returns observed for each root action during search. This addresses path degeneracy in the same manner but acts as a reduced variance estimator (as demonstrated in Figure 4, center, in the next section).

Modifications to MCTS's classic backpropagation step, such as TD-$\lambda$ (Sutton, 1988) variations, have been explored (Khandelwal et al., 2016). Such modifications are natural to incorporate into TSMCTS as well, especially with the aim to further reduce estimator variance. However, these have yet to popularize for MCTS, suggesting that they are not critical to the algorithm's performance and we leave their exploration in TSMCTS for future work. We include a brief algorithm summary of previous work on parallelizing MCTS and related challenges in Appendix A.5.

## 7 EXPERIMENTS

The objective of this work is to improve SMC as a search algorithm for policy improvement in RL with our novel method TSMCTS. To evaluate empirically that TSMCTS is a better policy improvement operator than SMC we use the experimental setup established by Macfarlane et al. (2024) and iterated upon by de Vries et al. (2025). This setup contains a mix of discrete and continuous control environments from Jumanji (Bonnet et al., 2024) and Brax (Freeman et al., 2021). de Vries et al. (2025) reduced the transition counts in evaluation to the standard in literature, and replaced one of the sparse-reward, single-goal environments (Boxoban) to a multi-reward environment (Snake), to increase the diversity of the environments covered in this experimental suite. We begin by comparing a model-based agent which uses **TSMCTS** for policy improvement (Algorithm 1) to other popular baselines which use search for policy improvement: **SPO** (Macfarlane et al., 2024), **TRT-SMC** (de Vries et al., 2025) and **GumbelAZ**, an AZ agent using a modern version of MCTS (Danihelka et al., 2022). All agents use the true dynamics model for search in the AZ manner. The SMC-based baselines (SPO, TRT SMC, TSMCTS) are agnostic to continuous / discrete action spaces. GumbelAZ has been extended to continuous environments in the manner of SampledMZ (Hubert et al., 2021). We include PPO (Schulman et al., 2017) for reference performance of a popular model-free baseline. Our implementation of all agents relies on that of de Vries et al. (2025), with the exception of SPO, which uses the original implementation (Toledo, 2024) in the environments for which it had been made public. As mentioned in Section 6, the contributions of de Vries et al. (2025) are for the most part orthogonal to ours. To demonstrate that this is the case, we include a **TSMCTS + TRT** agent which incorporates these contributions of de Vries et al. (2025) to the backbone of TSMCTS. See Appendices B and D for additional implementation details. The results are presented in Figure 1. In all environments the TSMCTS-based agents outperform or match all baselines.

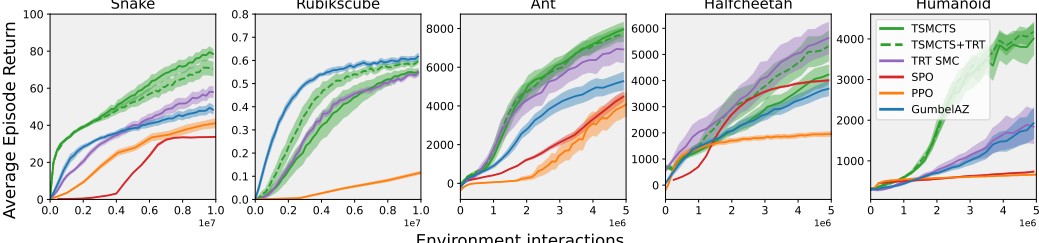

Figure 1: Averaged returns vs. environment interactions. 95% Gaussian CIs across 20 seeds.

We proceed to evaluate TSMCTS as a policy improvement operator directly. In Figure 2 we compare *identical* model based agents using the exact same implementation of Algorithm 1, differing only in the search procedure used for policy improvement: **TSMCTS**, the **SMC baseline** used by Macfarlane et al. (2024) and **GumbelMCTS**. We omit TRT SMC from this comparison as its modification to SMC have been shown to be orthogonal to TSMCTS's in Figure 1. This comparison also strengthens the connections between popular algorithmic setups of model based RL: the only difference between the GumbelMCTS agent, which is an AZ agent (GumbelAZ in Figure 1) and the SMC baseline, which is a simplified SPO agent (modified value targets, static temperature, etc.) is the search algorithm used for policy improvement. To emphasize the this connection we use the same colors for the related agents across figures. TSMCTS is overall the dominant search operator for policy improvement compared to both the SMC baseline and GumbelMCTS in these experiments.

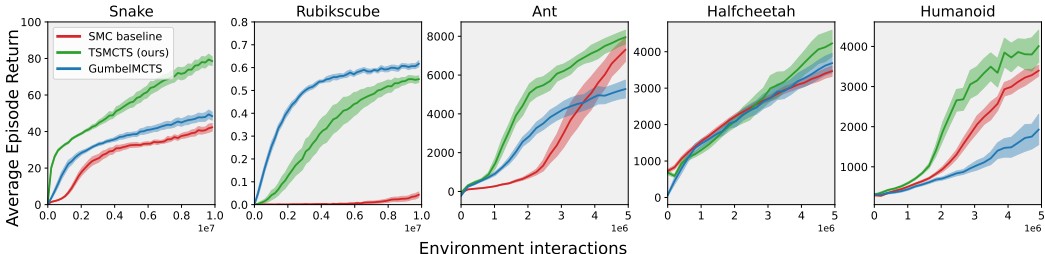

Figure 2: Averaged returns vs. environment interactions. 95% Gaussian CIs across 20 seeds.

In Figure 3 we include a reference runtime comparison between the three search algorithms. Runtime was estimated by multiplying training step with average runtime-per-step. TSMCTS induces a modest runtime increase over SMC for the same compute resources and compares very well to MCTS which has roughly twice the runtime cost as the SMC-based variants.

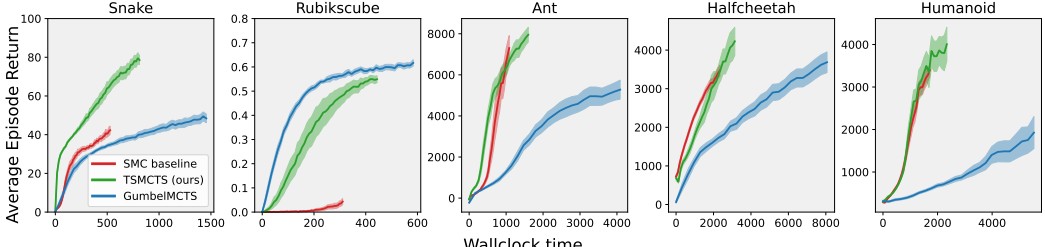

Figure 3: Averaged returns vs. runtime (seconds). Mean and 95% Gaussian CI across 20 seeds.

Next, we demonstrate empirically that TSMCTS addresses the limitations of SMC discussed in this work. In Figure 4 we plot: (i) Scaling with sequential compute (increasing depth $T$, left). (ii) Variance of the root estimator (center). (iii) Policy collapse at the root (target degeneracy) as a measure for path degeneracy (right).

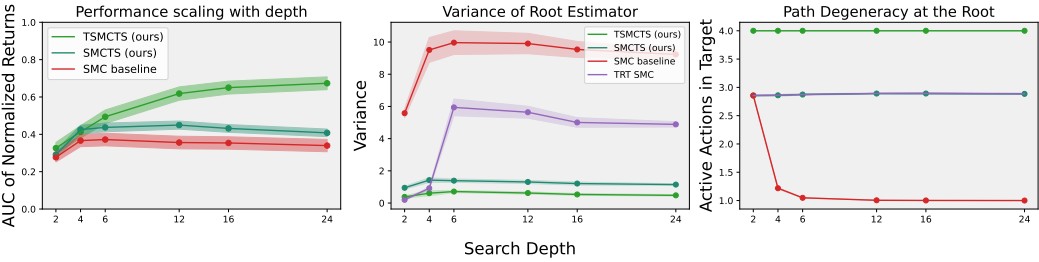

Figure 4: **Left:** Performance scaling with depth (higher is better), averaged across environments and particle budgets of $4, 8, 16$. 10 seeds and 90% two-sided BCa-bootstrap intervals. **Center:** Variance of the root estimator vs. depth (lower is better). **Right:** The number of actions active in the policy target (constant - no target degeneracy - better). Center and right are averaged across states and particle budgets $4, 8, 16$ and 5 seeds in Snake. Mean and $95\%$ Gaussian CI.

We compare baseline SMC, the intermediary SMCTS and the final algorithm TSMCTS. In the variance and path degeneracy experiments we include an SMC variant which uses the mechanism proposed by de Vries et al. (2025) for mitigating path degeneracy (e.g. TRT SMC). This, to demonstrate that while this mechanism mitigates path degeneracy in the same manner as SMCTS it does not address estimator variance as well. Performance is summarized as area-under-the-curve (AUC) for the evaluation returns during training normalized across environments. The normalization is with respect to minimum and maximum AUCs observed over all agents and seeds per environment. The variance measured is over the prediction of the root estimator for each planner $\mathbb{V}[V_{search}(s)] = \mathbb{V}[\sum_{a \in A} \pi_{improved}(a|s)Q_{search}(s,a)]$ (where $A$ is the set of actions searched by the respective search algorithm). The variance is computed across $L = 128$ independent calls to each planner per seed at every state in an evaluation episode after training has completed in the Snake environment, averaged across states and seeds. Target degeneracy is measured as the number of active actions in the policy target. The number of active actions at the root is averaged across the $L$ calls to the search algorithm.

TSMCTS is the only SMC variant to successfully scale with sequential compute (Figure 4 left). TSMCTS and SMCTS have significantly reduced estimator variance compared to the other SMC variants and TSMCTS's is significantly reduced compared to SMCTS's (Figure 4 center). All variants other than baseline address policy collapse at the root. TRT SMC and SMCTS however are limited by the entropy of the policy: the policy has high probability for only two actions in most states despite the size of the action space being 4 in this environment and thus only two actions are searched. TSMCTS on the other hand searches a constant $m_1 = 4$ actions, irrespective of the prior policy.

We investigate the effect of the hyperparameter $m_1$ of TSMCTS on the performance of the agent in Figures 5 in Appendix C. The effect appears overall marginal for sufficiently large $m_1 \geq 4$.

## 8 CONCLUSIONS

We presented Twice Sequential Monte Carlo Tree Search (TSMCTS), a search algorithm based in Sequential Monte Carlo (SMC) for action selection and policy optimization in Reinforcement Learning (RL). TSMCTS builds upon our formulation of SMC for search in RL which extends prior work (Piché et al., 2019) beyond the framework of Control As Inference (see Levine, 2018). TSMCTS harnesses mechanisms from Monte Carlo Tree Search (Świechowski et al., 2023) and Sequential Halving (Karnin et al., 2013) to mitigate the high estimator variance and path degeneracy problems of SMC, while maintaining SMC's beneficial runtime and space complexity properties. In experiments across discrete and continuous environments TSMCTS outperforms the SMC baseline as well as a popular modern version of MCTS (GumbelMCTS, Danihelka et al., 2022). In contrast to the SMC baseline, TSMCTS demonstrates lower estimator variance, mitigates the effects of path degeneracy at the root and scales favorably with sequential compute.

## REPRODUCIBILITY STATEMENT

Special care was taken to support reproducibility. Proofs and more detailed discussion of theoretical results are provided in Appendix A. Implementation details are described in Appendix B. Hyperparameters are listed in Appendix D. The codebase will be made public upon acceptance.

## LLM USAGE

LLMs were used in a minor role, to improve a small number of text paragraphs and for additional, supplementary, retrieval and discovery of related work.

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

## A THEORETICAL RESULTS

### A.1 RL-SMC IS A POLICY IMPROVEMENT OPERATOR

*Proof.* Given exact evaluation $Q^\pi$, true environment model $P, r$, a starting state $s_0$ and infinitely many particles $N \to \infty$, the SMC target policy at final step $T$ produces the following distribution over trajectories:

$$p(\tau_T) = p(s_0, a_0, \ldots, s_T, a_T, s_{T+1}) = \Pi_{i=0}^T P(s_{i+1}|s_i, a_i)\pi'(a_i|s_i) \tag{21}$$

The distribution $p(\tau_T)$ is equivalent to the distribution induced by following the policy $\pi'$ for all states $s_{0,\ldots,T}$, and for all other states following $\pi$, by definition. We call this policy $\pi_{SMC}$. We have:

$$V^\pi(s_0) \leq \mathbb{E}_{\pi'}[Q^\pi(s_0, a_0)] \tag{22}$$
$$= \mathbb{E}_{\pi',P}[r_0 + \gamma V^\pi(s_1)] \tag{23}$$
$$\leq \mathbb{E}_{\pi',P}[r_0 + \gamma Q^\pi(s_1, a_1)] \tag{24}$$
$$\leq \mathbb{E}_{\pi',P}[r_0 + \gamma r_1 + \gamma^2 Q^\pi(s_2, a_2)] \tag{25}$$
$$\leq \ldots \tag{26}$$
$$\leq \mathbb{E}_{\pi',P}[r_0 + \cdots + \gamma^{T-1}r_{T-1} + \gamma^T Q^\pi(s_T, a_T)] \tag{27}$$
$$= V^{\pi_{SMC}}(s_0) \tag{28}$$

Equation 22 holds by definition of $\pi'$ produced from an improvement operator. Note that actions $a_0, a_1, a_2, \ldots$ are all sampled from $\pi'(s_1), \ldots$ respectively, as the expectation is with respect to $\pi'$ at all steps. Equation 24 holds because $\mathbb{E}_{\pi'} Q^\pi(s_2, a) \geq V(s_2)$, by definition of $\pi'$. Equation 25 is the two-step expansion following the same argumentation, and respectively Equation 27 is the multi-step expansion, which is the definition of the value of the policy $\pi_{SMC}$.

$\square$

## A.2 DERIVING CAI-SMC IN RL-SMC

The importance sampling weights of CAI-SMC derive as follows (see the work of Piché et al. (2019)):

$$u_t(\tau_t \mid \tau_{t-1}) = P(s_t \mid s_{t-1}, a_{t-1}) \, \pi_\theta(a_t \mid s_t), \tag{29}$$

$$p_t(\tau_t \mid \tau_{t-1}) \propto P(s_t \mid s_{t-1}, a_{t-1}) \, \mu(a_t \mid s_t) \, \mathbb{E}_{s_{t+1} \mid s_t, a_t} \big[ \exp(A_{\text{soft}}(s_t, a_t)) \big], \tag{30}$$

$$w_t^n = w_{t-1}^n \frac{p_t(\tau_t^n \mid \tau_{t-1}^n)}{u_t(\tau_t^n \mid \tau_{t-1}^n)} \propto w_{t-1}^n \frac{\mu(a_t^n \mid s_t^n)}{\pi_\theta(a_t^n \mid s_t^n)} \mathbb{E}_{s_{t+1}^n \mid s_t^n, a_t^n} \big[ \exp(A_{\text{soft}}(s_t^n, a_t^n, s_{t+1}^n)) \big], \tag{31}$$

Denote:

$$\pi'(a_t^n \mid s_t^n) = \mu(a_t^n \mid s_t^n) \mathbb{E}_{s_{t+1}^n \mid s_t^n, a_t^n} \big[ \exp(A_{\text{soft}}(s_t^n, a_t^n, s_{t+1}^n)) \big] \tag{32}$$

Where $\pi'$ here is the posterior probability of CAI's graphical model, or the optimal soft-policy (Piché et al., 2019):

$$\pi'(a_t^n \mid s_t^n) = \mu(a_t^n \mid s_t^n) \mathbb{E}_{s_{t+1}^n \mid s_t^n, a_t^n} \big[ \exp(A_{\text{soft}}(s_t^n, a_t^n, s_{t+1}^n)) \big] \tag{33}$$

$$= \mu(a_t \mid s_t) \exp[\ln p(O_{t:T} \mid s_t, a_t) - \ln p(O_{t:T} \mid s_t)] \tag{34}$$

$$= p(a_t \mid s_t) p(O_{t:T} \mid s_t, a_t) / p(O_{t:T} \mid s_t) \tag{35}$$

$$= p(a_t \mid s_t, O_{t:T}) \tag{36}$$

We have:

$$w_t^n = \propto w_{t-1}^n \frac{\mu(a_t^n \mid s_t^n)}{\pi_\theta(a_t^n \mid s_t^n)} \mathbb{E}_{s_{t+1}^n \mid s_t^n, a_t^n} \big[ \exp(A_{\text{soft}}(s_t^n, a_t^n, s_{t+1}^n)) \big] = w_{t-1}^n \frac{\pi'(a_t^n \mid s_t^n)}{\pi_\theta(a_t^n \mid s_t^n)} \tag{37}$$

Which recovers RL-SMC.

## A.3 DERIVING THE VALUE UPDATE IN TSMCTS

In MCTS, the value $V_N(s_t)$ at each node $s_t$ equals the average of all returns $\frac{1}{N} \sum_{i=1}^{N} \sum_{k=0}^{T-1} \gamma^k r_{t+k}^i + \gamma^T v_\phi(s_{t+T}^i)$ observed through this node. This is because the variance of the estimator is expected to reduce with $1/N$, the number of visitations. This also holds in SMC, where for large $N$, the error behaves approximately Gaussian with variance proportional to $1/N$ (Chopin, 2004). For this reason, we rely on the same idea in TSMCTS.

At each iteration $i$ of TSMCTS the value estimate $Q_{SMCTS}^i(s_0, a)$ was computed using $N(i, a)$ particles per action, and thus, the contribution of this value estimate to the total average should be $N(i, a)$.

Equation 19 (provided below again for readability) formulates exactly this weighted average: it sums across the total number of iterations $\log_2 m_1$. For each iteration, it multiplies $Q_{SMCTS}^i(s_0, a)$ by the weight $N(i, a)$. Finally, it normalizes the sum by $\sum_{i=1}^{\log_2 m_1} N(i, a)$:

$$\forall a \in M_1: \quad Q_{SH}^i(s, a) = \frac{1}{\sum_{i=1}^{\log_2 m_1} N(i, a)} \sum_{i=1}^{\log_2 m_1} N(i, a) Q_{SMCTS}^i(s, a)$$

## A.4 VARIANCE REDUCTION

Throughout this work, we describe different mechanisms that reduce variance in TSMCTS compared to the SMC framework TSMCTS is built upon. In this section we will describe and motivate each mechanism in more detail. We begin with an overall motivation for variance minimization.

Variance minimization is a fundamental objective in statistical estimation, as the quality of an estimator is typically assessed through its mean squared error (MSE) (Geman et al., 1992). The MSE admits a standard decomposition into the squared bias and the variance,

$$\mathrm{MSE} = \mathrm{Bias}^2 + \mathrm{Var}.$$

While bias captures systematic deviation from the true quantity, variance reflects the sensitivity of the estimator to fluctuations in the data. Minimizing variance - without changing the bias - therefor reduces to minimizing estimation error. We proceed to describe each variance-reducing mechanism in chronological order.

**Backpropagation in SMCTS**   The running means $\bar{Q}_t(s_0, a_0)$ maintained through backpropagation in SMCTS decompose into:

$$\bar{Q}_t(s_0, a_0) = \frac{1}{t} \sum_{i=t}^{t} Q_i(s_0, a_0) = \sum_{i=1}^{N} w_t^i \mathbb{1}_{a_0^{(i)} = a_0} \sum_{j=0}^{t} \gamma^j r_j^i + \gamma^{t+1} V^{\pi_\theta}(s_{t+1}^i). \tag{38}$$

$\bar{Q}_t(s_0, a_0)$ is a reduced variance estimator compared to $Q_t$ for two reasons.

(i) Consider the bootstrapped return:

$$Q_t(s_0, a_0) = \sum_{j=0}^{t} \gamma^j r_j^i + \gamma^{t+1} V^{\pi_\theta}(s_{t+1}^i). \tag{39}$$

For any $h < t$, the estimator $Q_h(s_0, a_0)$ terminates earlier and bootstraps from $V^{\pi_\theta}$ sooner. Since extending the horizon from $h$ to $t$ replaces a single (deterministic) bootstrap term with additional random rewards and transitions, it introduces extra stochasticity. Consequently, $\mathrm{Var}(Q_h(s_0, a_0)) < \mathrm{Var}(Q_t(s_0, a_0))$, reflecting the classical result that Monte Carlo returns (large $t$) have higher variance than temporally shorter, bootstrapped estimates (small $h$) (Sutton & Barto, 2018).

(ii) Let us assume for a moment the policy, transition dynamics and reward are all deterministic. Any errors in the value prediction $v_\phi$ that are I.I.D. will average out in the empirical average $\frac{1}{N} \sum_{t=1}^{N} \sum_{i=0}^{t} \gamma^i r_i + v_\phi(s_{i+1})$ where $s_i = P(s_{i-1}, \pi(s_{i-1}))$. For that reason mixing different length bootstrapped returns can result in reduced variance estimates compared to any individual bootstrapped return even when the dynamics and rewards are deterministic.

**Repeatedly searching the same actions from the root in TSMCTS**   At each iteration $i$, TSMCTS searches a set of actions $A_i \subset A_{i-1}$. Since the actions are searched independently again from the root, we have $\mathrm{Var}(Q_{SH}^i) < \mathrm{Var}(Q_{SMCTS}^i)$. That is, the average *across* the value estimates of independent iterations is a lower variance estimate of the true value compared to each individual estimate, for the same reasoning as above.

**Increasing particle budget per searched action at the root in TSMCTS**   Under standard assumptions, increasing the number of particles in SMC algorithms reduces variance because the particle system provides an empirical average, and the variance of such Monte Carlo estimates decreases proportionally to the number of particles $1/N$, where $N$ is the number of particles (Chopin & Papaspiliopoulos, 2020).

**Searching for a shorter horizon**   TSMCTS trades off the depth of the search $T_{SH} < T$ for repeated search from the root. Reducing the depth of the search has two main effects: (i) It reduces the number of consecutive improvement (or search) steps. In a manner of speaking, the resulting policy is "less improved". (ii) It results in a lower variance estimator, as the variance grows in $t$ and $T_{SH} < T$.

### A.5   COMPLEXITY ANALYSIS

We include a brief runtime and space complexity analysis for MCTS and RL-SMC.

**MCTS complexity**   For a search budget $B$, MCTS conducts $B$ iterations. At each iteration $i$, MCTS conducts $d_i \leq B$ search steps, one expansion step, and then $d_i \leq B$ backpropagation steps along the nodes in the trajectory. $d_i$ denotes the depth of the leaf at step $i$. We can therefor bound the runtime complexity by $\mathcal{O}(B(B + B + 1)) = \mathcal{O}(B^2)$ operations. In regards to space complexity, MCTS construct a tree of size $B$, so the space required is of complexity $\mathcal{O}(B)$.

**RL-SMC complexity**  For $N$ particles and a depth $T$, the search budget of RL-SMC totals $NT = B$. Assuming that $N$ particles operate in parallel the (sequential) runtime complexity is $\mathcal{O}(T) \leq \mathcal{O}(B) < \mathcal{O}(B^2)$ operations. In terms of space, RL-SMC maintains only statistics about each particle, resulting in space complexity of $\mathcal{O}(N) \leq \mathcal{O}(B)$. Since RL-SMC is merely a generalization of Piché et al. (2019)'s CAI-SMC, we conclude that CAI-SMC has the same space and runtime complexity.

**SMCTS complexity**  For $N$ particles and a depth $T$, the search budget of SMCTS totals $NT = B$. At each step $i$, SMCTS conducts a constant number of additional operation: one running sum is maintained for each particle, and one running sum is maintained for each searched action at the root. As a result, SMCTS maintains the same (sequential) runtime complexity of RL-SMC of $\mathcal{O}(T) < \mathcal{O}(B^2)$. SMCTS maintains statistics about $N$ particles, and also statistics about $M \leq N$ searched actions at the root. This results in space complexity of $\mathcal{O}(2N) = \mathcal{O}(N) \leq \mathcal{O}(B)$, the same space complexity as RL-SMC.

**TSMCTS complexity**  For $N$ particles and a depth $T$, the search budget of SMCTS totals $NT = B$. TSMCTS divides this budget across $\log_2 m_1$ iterations. At each iteration, TSMCTS executes $T/\log_2 m_1$ steps, resulting in runtime complexity of $\mathcal{O}(\log_2 m_1 \frac{T}{\log_2 m_1}) = \mathcal{O}(T) < \mathcal{O}(B^2)$, the same as SMCTS and RL-SMC. In terms of space complexity, TSMCTS maintains statistics over $N$ particles, and $m_1 \leq N$ searched actions at the root, resulting in the same space complexity as RL-SMC and SMCTS, $\mathcal{O}(2N) = \mathcal{O}(N) \leq \mathcal{O}(B)$.

**Parallelizing MCTS**  Approaches to parallelize MCTS exist (Chaslot et al., 2008). These range from running *leaf parallelization* which performs multiple independent rollouts from the same newly expanded leaf node, improving evaluation accuracy but not accelerating tree growth. This of course is not applicable with modern MCTS methods which use a value DNN to expand leaves. *Search parallelization* runs MCTS in parallel across multiple states in multiple environments in parallel. This is the current norm for JAX based implementations, such as by DeepMind et al. (2020). *Root parallelization* launches multiple independent MCTS instances — each constructing its own search tree — and aggregates root-level statistics. This is in in direct competition over resources with *search parallelization*. Since it runs multiple trees for the same state, it reduces the number of independent states that can be searched in parallel, and thus slows down data gathering (number of environment interactions per search steps). *Tree parallelization* is the most akin to the parallelization of SMC: it shares a single MCTS tree among multiple workers, requiring synchronization mechanisms—such as local mutexes and virtual loss—to maintain consistency and avoid redundant exploration. Overall, this contrasts very clearly with the ease at which SMC parallelizes. In SMC one can simply increase the number of particles $N$.

**A note on complexity in practice**  It is unlikely that all operations will have the same compute cost in practice. In search algorithms that use DNNs, it is often useful to think of two separate operation costs: model interactions, and DNN forward passes. Either of the two can often be the compute bottleneck, depending on the choice of model, DNN architecture, hardware etc. This motivates an equating for compute estimated in number of model expansions / DNN forward passes which is $B$ for MCTS and $NT$ for SMC, which is why we as well as previous work opted to compare MCTS and SMC variants with budgets $B = NT$.

# B  IMPLEMENTATION DETAILS

**Targets and losses**  Our implementation for all search-based agents uses a $v_\phi$ critic and a prior policy $\pi_\theta$. The value and policy are trained with the following losses:

$$\mathcal{L}(\theta) = \mathbb{E}_{(s_t, a_t, \pi_t) \sim \mathcal{D}_{(n)}} \left[ -\mathbb{E}_{a \sim \pi_t} \ln \pi_\theta(a|s_t) - c_{ent} \mathcal{H}[\pi_\theta(a|s_t)] \right], \tag{40}$$

$$\mathcal{L}(\phi) = \mathbb{E}_{(s_t, a_t, v_t) \sim \mathcal{D}_{(n)}} \left[ (v_t - v_\phi(s_t))^2 \right]. \tag{41}$$

$\pi_t$ is the policy target for state $s_t$ which is the policy $\pi_{improved}$ returned by the planner (be it SMC, SMCTS, TSMCTS or MCTS). $v_t$ is the value target for state $s_t$ which is computed using TD-$\lambda$ with bootstraps $V_{search}$ returned by the planner. $\mathcal{H}[\pi_\theta] = -\mathbb{E}_{\pi_\theta} \ln \pi_\theta$ is an entropy penalty for the policy. $\mathcal{D}_{(n)}$ is the replay buffer for iteration $n$.

**The training loop** The RL training setup follows the popular approach in JAX, which gathers interaction trajectories of length *unroll length* $L$ for a *batch size* $B$ in parallel, resulting in a total replay buffer of size $LB$ per episode. The agent is then trained for $K$ *SGD update steps* with *SGD minibatch size* (see hyperparameters in Table 1) and the above losses. Following that, the agent proceeds to gather a additional data of size $LB$. The AdamW optimizer (Loshchilov & Hutter, 2019) was used with an $l_2$ penalty of $10^{-6}$ and a learning rate of $3 \cdot 10^{-3}$. Gradients were clipped using a max absolute value of 10 and a global norm limit of 10.

**Discrete vs. continuous action spaces** The same losses are used to train the value and policy networks, irrespective of the type of action space. In continuous environments, the policy is a Gaussian policy, predicting mean and variance. In discrete environments, the policy is trained to predict the log-probabilities for each action in the action space, as is standard, using the empirical cross entropy loss $\mathcal{L}(\theta)$.

MCTS was originally designed for discrete action environments, and is slightly less agnostic to continuous actions. We follow the popular approach of SampledMZ (Hubert et al., 2021), which showed that one can simply sample $K$ actions from the prior policy at each node in the search tree, and treat $\{a_1, \ldots, a_K\}$ is a discrete action space, turning MCTS into a continuous-action-space algorithm.

Pseudocode for the different algorithms is provided below.

---

**Algorithm 1** Outer-Loop with Modular Search

---

**Require:** Search algorithm (planner) $\mathcal{P}$, neural networks $\pi_{\theta_1}, V_{\phi_1}$, replay buffer $\mathcal{D}_{(1)} = \emptyset$, environment's dynamics model $\mathcal{M} = (P, R)$ and budget parameters $B$.

1: **for** episode $n = 1$ to $N$ **do**
2:      Sample starting state $s_1 \sim \rho$.
3:      **for** step $t = 0$ to termination or timeout **do**
4:          $\pi_{improved}(s_t), V_{search}(s_t) \leftarrow \mathcal{P}(\pi_{\theta_n}, v_{\phi_n}, \mathcal{M}, B)(s_t)$.
5:          $a_t \sim \pi_{improved}(s_t)$.
6:          $s_{t+1} \sim P(\cdot|s_t, a_t), \quad r_t \sim R(s_t, a_t)$.
7:          Append $(s_t, a_t, r_t, s_{t+1}, \pi_{improved}(s_t), V_{search}(s_t))$ to buffer $\mathcal{D}_{(n)}$.
8:      **end for**
9:      Update policy params $\theta_{n+1}$ with SGD and CE loss on targets $\pi_{improved}$ from $\mathcal{D}_{(n)}$.
10:     Update value params $\phi_{n+1}$ with SGD and MSE loss on TD-$\lambda$ targets using $V_{search}$ from $\mathcal{D}_{(n)}$.
11:     Set $\mathcal{D}_{n+1} = \mathcal{D}_n$.
12: **end for**

---

**Algorithm 2** RL-SMC

---

**Require:** Number of particles $N$, depth $T$, model $P$, prior-policy $\pi_\theta$, policy improvement operator $I$, value function $Q^{\pi_\theta}$ and current state in the environment $s_{root}$.

1: Initialize particles $n \in N$, with $w_0^n = 1, s_1^n = s_{root}$, and ancestor identifier $\{j_1^n = n\}_{n=1}^N$ (which identifies per particle which action at the root it is associated with).
2: **for** $t = 1$ to T **do**
3:      *Mutation*: $\{a_t^n \sim \pi_\theta(a_t|s_t^n)\}_{n=1}^N, \quad \{s_{t+1}^n \sim P(\cdot|s_t^n, a_t^n)\}_{n=1}^N$.
4:      *Correction*: $\{w_t^n = w_{t-1}^n \frac{\pi'(a_t^n|s_t^n)}{\pi_\theta(a_t^n|s_t^n)}, \quad \pi'(s_t^n) = \mathcal{I}(Q^{\pi_\theta}, \pi_\theta)(s_t^n)\}_{n=1}^N$.
5:      *Selection*: $\{(j_t^n, a_t^n, s_{t+1}^n)\}_{n=1}^N \sim \text{Multinomial}(N, \text{normalized } w_t), \quad \{w_t^n = 1\}_{n=1}^N$.
6: **end for**
7: Return $\{j_T^n, w_T^n\}_{n=1}^N$

---

---

**Algorithm 3** SMCTS

---

**Require:** Number of particles $N$, depth $T$, current state in the environment $s_{root}$, model $\mathcal{M} = (P, R)$, prior-policy $\pi_\theta$, value network $v_\phi$, improvement operators for search $\mathcal{I}_{search}$ and root $\mathcal{I}_{root}$.

1: Initialize particles $n \in N$, with $w_0^n = 1, s_1^n = s_{root}$ non-bootstrapped returns $R_0^n = 0$, and ancestor identifier $\{j_1^n = n\}_{n=1}^N$.

2: **for** $t = 1$ to $T$ across $n$ particles in parallel **do**

3:     *Mutation*: $a_t^n \sim \pi_\theta(s_t^n), r_t^n \sim R(s_t^n, a_t^n), s_{t+1}^n \sim P(s_{t+1}^n | s_t^n, a_t^n)$.

4:     If $t = 1$, maintain the set of $N$ root actions: $A_1 \leftarrow \{a_1^n\}_{n=1}^N$.

5:     Approximate state-action value: $Q(s_t^n, a_t^n) \leftarrow r_t^n + \gamma v_\phi(s_{t+1}^n)$.

6:     Compute the search policy: $\pi'(s_t^n) \leftarrow \mathcal{I}_{search}(\pi_\theta, Q)(s_t^n)$.

7:     *Correction:* Compute importance sampling weights (Equation 11): $w_t^n = w_{t-1}^n \frac{\pi'(s_t^n)}{\pi_\theta(s_t^n)}$.

8:     Update the non-bootstrapped returns: $R_t^n = R_{t-1}^n + \gamma^t r_t^n$.

9:     Normalize importance sampling weights *per action at the root* using their identifiers $j_t^n$:

$$y_t^n = \frac{w_t^n}{\sum_{k=1}^N w_t^j \mathbb{1}_{j_t^n = j_t^k}}.$$

10:     Estimate $Q_t(s_{root}, \cdot)$ for initial actions $a_1^n \in A_1$ using $R_t^n$ and $v_\phi(s_{t+1})$:

$$Q_t(s_{root}, a_1^n) = \sum_i^N y_t^i (R_t^i + \gamma^{t+1} v_\phi(s_{t+1}^i)) \mathbb{1}_{j_1^n = j_t^i}$$

11:     Update the running average $\bar{Q}_t(s_{root}, \cdot)$ where $Q_t(s_{root}, \cdot)$ is defined:

$$\bar{Q}_t(s_{root}, a_1^n) = \frac{(1 - t)\bar{Q}_{t-1}(s_{root}, a_1^n) + Q_t(s_{root}, a_1^n)}{t}$$

12:     *Selection:* Resample particles proportional to $w_t$, and reset $w_t \leftarrow 1$, as in Algorithm 2.

13: **end for**

14: Compute improved policy $\pi_{improved}(s_{root}) \leftarrow \mathcal{I}_{root}(\pi_\theta, \bar{Q}_T)(s_{root})$.

15: Compute the value of the improved policy across the set of root actions $A_1$:

$$V_{search}(s_{root}) \leftarrow \sum_{a_1^n \in A_1} Q_T(s_{root}, a_1^n) \pi_{improved}(a_1^n | s_{root}).$$

16: Return $\pi_{improved}(s_{root}), V_{search}(s_{root})$.

---

**Algorithm 4** TSMCTS

---

**Require:** Number of particles $N$, planning depth $T$, current state in the environment $s_{root}$, number of actions to search at the root $m_1$, model $\mathcal{M} = (P, R)$, policy network $\pi_\theta$, value network $v_\phi$, Gumbel noise vector $g$.

1: Compute the per-iteration depth (Equation 17): $T_{SH} \leftarrow T / \log_2 m_1$.

2: Get $m_1$ starting actions (Equation 18, left): $A_1 = \{a_1, \ldots, a_{m_1}\} \leftarrow \arg\text{top}(\pi_\theta(s_{root}) + g, m_1)$.

3: Initialize running sum of particles per action $N^0(a) \leftarrow 0$ and running value sum per action $Q^0_{sum}(s_{root}, a) \leftarrow 0$, for all actions at the root $a \in A_1$.

4: Compute starting number of particles per action: $N_1 \leftarrow floor(N/m_1)$.

5: **for** $i = 1$ to $\log_2 m_1$ **do**

6:     **for** each action $a \in A_i$ in parallel **do**

7:         Sample $s_1 \sim P(\cdot|s_{root}, a)$, $r_1 \sim R(s_{root}, a)$.

8:         Search using SMCTS:

$$\_, V^i_{SMCTS}(s_1) \leftarrow \text{SMCTS}(N_i, T_{sh}, s_1, \mathcal{M}, \pi_\theta, v_\phi, \mathcal{I}_{GMZ}, \mathcal{I}_{GMZ}).$$

9:         Approximate the value of action $a$:

$$Q^i_{SMCTS}(s_{root}, a) = r + \gamma V_{SMCTS}(s_1).$$

10:        Update the running sums of particles and values of $a$ (Equation 20):

$$N^i(a) \leftarrow N^{i-1}(a) + N_i, \quad Q^i_{sum}(s_{root}, a) \leftarrow Q^{i-1}_{sum}(s_{root}, a) + N_i Q^i_{SMCTS}(s_{root}, a).$$

11:     **end for**

12:     Compute the current iteration's value estimate at the root (Equation 19):

$$\forall a \in A_i: \quad Q^i_{SH}(s_{root}, a) \leftarrow \frac{1}{\sum_{j=1}^i N(j, a)} \sum_{j=1}^i N(i, a) Q^j_{SMCTS}(s_{root}, a) = \frac{Q^i_{sum}(s_{root}, a)}{N^i(a)}.$$

13:     Update the number of actions to search: $m_{i+1} = m_i / 2$.

14:     Update the actions to search (Equation 18, right): $A_{i+1} \leftarrow \arg\text{top}(Q^i_{SH}(s_{root}, \cdot), m_{i+1})$

15:     Update the running number of particles per action: $N_{i+1} \leftarrow 2N_i$.

16: **end for**

17: Compute the final Q-estimate (Equation 19):

$$\forall a \in A_1: \quad Q_{SH}(s_{root}, a) \leftarrow \frac{Q^{\log_2 m_1}_{sum}(s_{root}, a)}{N^{\log_2 m_1}(a)}.$$

18: Compute the improved policy $\pi_{improved}$ using $\mathcal{I}_{GMZ}$:

$$\forall a \in A_1: \quad \pi_{improved}(a|s_{root}) \leftarrow \frac{\exp(\beta Q_{sh}(s_{root}, a) + \log \pi_\theta(a|s_{root}) + g(a))}{\sum_{b \in A_1} \exp(\beta Q_{sh}(s_{root}, b) + \log \pi_\theta(b|s_{root}) + g(b))}$$

$$\forall a \notin A_1: \quad \pi_{improved}(a|s_{root}) \leftarrow 0$$

19: And its value: $V_{search}(s_{root}) = \sum_{a \in M_1} \pi_{improved}(a|s_{root}) Q_{sh}(s_{root}, a)$.

20: Return $\pi_{improved}(s_{root})$, $V_{search}(s_{root})$.

---

## C  ADDITIONAL EXPERIMENTS

**Investigating the effect of the $m_1$ parameter**  We investigate the effect of the $m_1$ parameter, the number of actions that are searched at the root of TSMCTS in Figure 5. Performance is summarized as area-under-the-curve (AUC) for the evaluation returns during training normalized across environments with respect to minimum and maximum AUCs observed across agents and seeds. Clearly, limiting the search to only the top two actions is strongly detrimental. On the other hand the confidence bounds for $m_1 = 4, 16$ almost entirely overlap, which suggest that for a sufficiently large $m_1$ the effect across environments is modest.

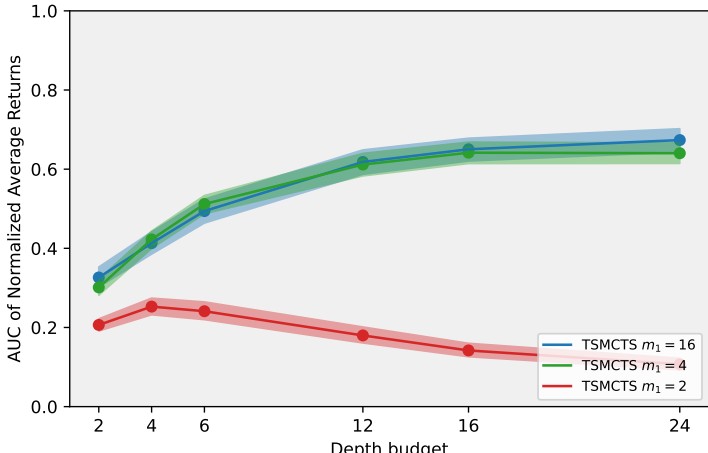

Figure 5:  Performance scaling with depth (higher is better, increasing is better). Averaged across environments and particle budgets of $4, 8, 16$ and normalized across environments. Mean and 90% two-sided BCa-bootstrap intervals across 10 seeds.

## D  EXPERIMENTS DETAILS

For the experiments, we build on the setup proposed by de Vries et al. (2025), which we describe in more detail below.

**Environments**  We have used Jumanji's (Bonnet et al., 2024) Snake-v1 and Rubikscube-partly-scrambled-v0, as well as Brax's (Freeman et al., 2021) Ant, Halfcheetah and Humanoid.

**Compute**  All experiments were run on the [anonymized for review] cluster with a mix of [anonymized for review] GPU cards. Each individual run (seed) used 2 CPU cores and $\leq 6$ GB of VRAM.

**Wall-clock Training Time Estimation**  To estimate the training runtime in seconds (Figure 3), we used an estimator of the the runtime-per-step (total runtime divided by steps) and multiplied this by the current training step to obtain a cumulative estimate. This estimator should more robustly deal with the variations in hardware, the compute clusters' background load and XLA dependent compilation. Of course, estimating runtime is strongly limited to hardware and implementation and the results presented in Figure 3 should only be taken with that in mind.

**Variance and Path Degeneracy Estimation**  In Figure 4 center we plot the variance of the root estimator $V(s_0) = \sum_{a \in M} \pi_{improved}(a|s)Q_{search}(s, a)$ at the end of training as a function of depth for each method. $M$ is the number of actions over which the estimator maintains information (susceptible to path degeneracy).

Following de Vries et al. (2025), for TRT-SMC and the SMC baseline we compute $Q_{search}$ as the TD-$\lambda$ estimator for each particle at the root for the *last* depth $t = T$. If multiple particles are associated with the same action at the root, the particle estimates are averaged. To address path

degeneracy when all particles for a root action are dropped TRT-SMC saves the last TD-$\lambda$ estimate for each root action. For T/SMCTS we use $V_{T/SMCTS}$ respectively. In Figure 4 right we plot the number of actions at the root with which information is associated at the end of training, $M$, vs. depth.

**Neural Network Architectures**   As specified by de Vries et al. (2025), which are themselves adapted from Bonnet et al. (2024) and Macfarlane et al. (2024) (e.g. MLPs in all environments except Snake where a CNN followed by an MLP is used).

**Hyperparameters**   We've used the hyperparameters used by Macfarlane et al. (2024) and de Vries et al. (2025) for these tasks (when conflicting, we've used the parameters used by the more recent work (de Vries et al., 2025)). Except for the two new hyperparameters introduced by T/SMCTS no hyperparameter optimization took place. These new hyperparameters are (i) $m_1$, for which results are presented in Figure 5. (ii) The $\beta_{root}$ inverse-temperature hyperparameter of $\mathcal{I}_{GMZ}$ used by T/SMCTS to compute the improved policy at the root ($\mathcal{I}_{root}$). For $\beta_{root}$ we conducted a grid search with a small number of seeds across environments and values of $0.1^{-1}, 0.05^{-1}, 0.01^{-1}, 0.005^{-1}$. $\beta = 0.01^{-1}$ was overall the best performer. The $\beta_{search}$ hyperparameter is actually the same parameter as the *target temperature* used by the SMC baseline (see de Vries et al., 2025). We have not observed differences in performance across a range of parameters $\beta_{search}$ for TSMCTS and opted to use the same value as SMC.

Hyperpameters are summarized in Tables 1, 2, 3, 4, 5 and 6.

| Name | Value Jumanji | Value Brax |
|---|---|---|
| SGD Minibatch size | 256 | 256 |
| SGD update steps | 100 | 64 |
| Unroll length (nr. steps in environment) | 64 | 64 |
| Batch-Size (nr. parallel environments) | 128 | 64 |
| (outer-loop) Discount | 0.997 | 0.99 |
| Entropy Loss Scale ($c_{ent}$) | 0.1 | 0.0003 |

Table 1: Shared experiment hyperparameters.

| Name | Value Jumanji | Value Brax |
|---|---|---|
| Policy-Ratio clipping | 0.3 | 0.3 |
| Value Loss Scale | 1.0 | 0.5 |
| Policy Loss Scale | 1.0 | 1.0 |
| Entropy Loss Scale | 0.1 | 0.0003 |

Table 2: PPO hyperparameters.

| Name | Value Jumanji | Value Brax |
|---|---|---|
| Replay Buffer max-age | 64 | 64 |
| Nr. bootstrap atoms | 30 | 30 |
| Max depth | 16 | 16 |
| Max breadth | 16 | 16 |

Table 3: GumbelMCTS hyperparameters.

| Name | Value Jumanji | Value Brax |
|---|---|---|
| Replay Buffer max-age | 64 | 64 |
| Selection (Resampling) period | 4 | 4 |
| Target temperature | 0.1 | 0.1 |
| Nr. bootstrap atoms | 30 | 30 |

Table 4: Shared SMC hyperparameters.

| Name | Value Jumanji | Value Brax |
|------|---------------|------------|
| (inner-loop) Retrace $\lambda$ | 0.95 | 0.9 |
| (inner-loop) Discount | 0.997 | 0.99 |
| (outer-loop) Value mixing | 0.5 | 0.5 |
| Estimation $\pi_{improved}$ | Message-Passing | Message-Passing |

Table 5: TRT-SMC variance ablation hyperparameters.

| Name | Value |
|------|-------|
| Root policy improvement operator ($\mathcal{I}_{root}$) | $\mathcal{I}_{GMZ}$ |
| Search policy improvement ($\mathcal{I}_{search}$) | $\mathcal{I}_{GMZ}$ |
| Root inverse temperature $\beta_{root}$ | $0.01^{-1}$ |
| Search inverse temperature $\beta_{search}$ | $0.1^{-1}$ |
| Number of actions to search at the root $m_1$ | 4 (Figures 1,2,3), 16 (Figure 4) |

Table 6: SMCTS and TSMCTS hyperparameters.

