# OpenReview forum: "Parallelizing Tree Search With Twice Sequential Monte Carlo"
_ICLR.cc/2026/Conference — Submitted to ICLR 2026_

### Official Review · Reviewer_bDyy · 2025-10-28

**Soundness:** 3
**Presentation:** 3
**Contribution:** 3
**Rating:** 6
**Confidence:** 2

**Summary:**

The paper focuses on model-based reinforcement learning (MBRL) methods that leverage search algorithms.
Specifically, the paper claims that Monte-Carlo Tree Search (MCTS) does not fully utilize the GPU when compared to Sequential Monte Carlo (SMC).
However, the latter suffers from high variance and path degeneracy.
As a result, the paper proposes a method that combines the two, called Twice Sequential Monte-Carlo Tree Search (TSMCTS) that aims to improve upon both existing algorithms.
The paper demonstrates through both discrete and continuous environments TSMCTS is comparable if not better than existing algorithms in episodic returns, and further demonstrates that TSMCTS maintains smaller variance and less path degeneracy.

**Strengths:**

- The proposed method clearly yields better runtime while maintaining/improving performance when compared to Gumbel MCTS and SMC.
- The proposed method suffers less from high variance and path degeneracy.
- The complexity analysis in the appendix A.4 is helpful to understand the asymptotic trade-offs.

**Weaknesses:**

- Method
	- On line 268, keeping track of $t$ Q-values can be prohibitively expensive---instead something akin to $Q(\lambda)$ might just do the work.
	- It seems like sequential halving can suffer from high bias if the first iteration filters out good action early---this can easily be the case with stochastic dynamics and low search budget. If this is true, I think the paper should indicate this, otherwise it would be nice to provide a reference, demonstrating that this is not the case.
- Writing
	- On line 116, the notation $s_t$ conflates with the timesteps in the MDP. I suggest disambiguating them with, e.g., superscript, $\bar{s}_t$, etc.
	- Nit: On line 187, I think it should be $\tau_t = s_0, a_0, \dots, s_t, a_t, s_{t + 1}$
	- Theorem 1 should indicate that the assumptions are true $Q^\pi$ and transitions $P$, otherwise it can be misleading in the function-approximation case.
	- I think the writing can be clearer on line 234. If I understand correctly, $a_0^i \sim \pi_\theta(\cdot | s_0)$ means that most particles will start from the most-likely action, and consequently the empirical distribution will only have mass on said action. On the other hand, MCTS uses the estimated Q-value to form the empirical distribution.
- Experiments
	- The experiments should describe how the continuous environments are setup---is the action space discretized, or the cross-entropy loss is replaced with a squared loss, etc.?
	- How does this approach compare against Levin Tree Search and Luby Tree Search [1]?
	- This doesn't affect my score, but how would it perform when the transitions are approximated?

References:
[1] Orseau, Laurent, et al. "Single-agent policy tree search with guarantees." Advances in Neural Information Processing Systems 31 (2018).

**Questions:**

See above

---

> ### Author Response · Authors · 2025-11-15
>
> Dear reviewer,
>
> Thank you for your review and many valuable comments and questions.
>
> **Method**:
>
> 1. Note that we do not keep track of $t$ Q values, only the sum. So the cost is always $\mathcal{O}(min(N, m_1))$, where $N$ is number of particles and $m_1$ is the number of actions that are searched at the root. However, we agree with the reviewer that TD-$\lambda$ is a promising direction for future work (see second paragraph of Related Work).
>
> 2. This is not the case: neither SMC nor SH have error in expectation (i.e. neither is biased).
> While SH can eliminate good actions by chance, so can SMC. In addition, the probabilities of eliminating good actions are bounded.
> Note that when SMC (/SMCTS) is used, with finite depth $T$ and finite particles $N$, *for each action at the root independently in parallel* (as we use for SH in TSMCTS), each "arm" (each action at the root) is a stationary bandit.
> For that reason, the "almost-optimal"ity guarantees of SH for best-action identification (see [1]) hold for TSMCTS.
>
> **Writing:**
>
> Thank you for the valuable comments.
> We have incorporated the comments into the edit uploaded with this rebuttal (edits marked in green).
>
> **Experiments:**
>
> 1. Thank you for pointing out that this was not sufficiently clear.
> The SMC variants are agnostic to continuous / discrete actions: they use an empirical cross-entropy loss with the logits of the particles / actions at the root (e.g. the improved policy).
> The MCTS variant uses SampledMZ's [2] approach (which is essentially the same, except that during search the action space is "discretized" by sampling some $K$ actions from the prior policy, and treating them as a discrete action space).
> We have clarified this in Appendix B in a dedicated paragraph (edits in green).
>
> 2. The objective of this paper is to improve SMC-based search for policy improvement.
> We include MCTS for reference, as a popular modern search algorithm used for this purpose across a wide range of domains and literature.
> The methods in the work referred to by the reviewer (Orseau et al.) are not used for policy improvement, and similarly are not compared with RL methods such as AlphaZero [3]. For that reason, we have not included them in the comparison.
>
> 3. [4] applied SMC successfully on a learned model with approximate transitions.
> As TSMCTS iterates on the search developed by this prior work, we would expect it to work well, as well.
>
> If any questions or concerns remain unaddressed, please let us know.
>
> References:
>
> [1] Karnin et al. "Almost Optimal Exploration in Multi-Armed Bandits". ICML 2013.
>
> [2] Hubert, Thomas, et al. "Learning and planning in complex action spaces." ICML, 2021.
>
> [3] Silver et al. "Mastering Chess and Shogi by Self-Play with a General Reinforcement Learning Algorithm." Science, 2018.
>
> [4] Piché, Alexandre, et al. "Probabilistic planning with sequential monte carlo methods." ICLR 2018.

---

### Official Review · Reviewer_oJ94 · 2025-10-31

**Soundness:** 1
**Presentation:** 2
**Contribution:** 2
**Rating:** 2
**Confidence:** 5

**Summary:**

This paper proposes TSMCTS, a new Sequential Monte Carlo (SMC) search algorithm for policy improvement. The method is presented as an alternative to MCTS that aims to solve the high variance and path degeneracy issues found in standard SMC for RL methods. The authors build their method incrementally: first, by formalising SMC beyond the Control-as-Inference (CAI) framework; second, by introducing SMCTS, which adds MCTS-inspired value backpropagation to mitigate path degeneracy; and finally, by integrating Sequential Halving at the root to create TSMCTS for better budget allocation. The paper's experiments show TSMCTS outperforming a baseline SMC and GumbelMCTS, claiming it yields lower-variance targets and scales properly with the sequential budget.

**Strengths:**

This paper's focus is particularly impactful as it targets two fundamental weaknesses that hinder the scalability and performance of Sequential Monte Carlo (SMC) in Reinforcement Learning:

* Targeting **path degeneracy** by preserving information about multiple actions at the root is a crucial research direction. This prevents the search from prematurely collapsing to a single trajectory, which leads to unstable and impoverished policy targets—a key limitation of standard SMC.
* Developing systematic **variance reduction** mechanisms is vital for making SMC a scalable planning algorithm. Focusing on techniques that combat the high variance growth with search depth is a key direction, as this is a fundamental bottleneck that prevents SMC from benefiting from deeper lookaheads.

**Weaknesses:**

### 1. Insufficient Baselines
* **Ambiguous "SMC Baseline":** The paper introduces an "SMC baseline" that doesn't correspond to a specific, recognized algorithm in the literature. This is confusing and unscientific. The existing methods are distinct: the SMC method from Piché et al. [3], SPO (which uses SMC for policy improvement) [2], and the extension by de Vries et al. with TRT-SMC (SPO + twisted proposals + revived sampling)[4]. The authors should compare against each of these established methods in all experiments. In addition, baselines such as SPO have open-source implementations, so their omission is a significant weakness of this paper. These baselines are a strict requirement, and I would strongly push for rejection without this being fully addressed.
### 2. Flawed Narrative and Positioning
* **Insufficient Coverage of Prior Work:** The related work section incorrectly frames the history of SMC in RL. It misses the seminal work by Lazaric et al. [1] that first introduced SMC for RL. The progression should be: Lazaric et al. [1] -> Piché et al. -> SPO (full policy iteration loop, highlighting parallelism) -> de Vries et al. (improvements on SPO via SMC and proposal changes).
* **Overstated Claims on Parallelism:** The title and framing imply that this paper is the first to identify the parallelism benefits of SMC for RL. This benefit was already robustly established and was a critical contribution of SPO. It is fine to reference this as a benefit of SMC-type methods, but the correct credit must be assigned to SPO where this was demonstrated.
* **Unclear Motivation for Decoupling from Control as Inference (CAI):** The paper doesn't provide a clear rationale for moving away from the CAI framework, especially since SMC is fundamentally a probabilistic inference method for which CAI is a natural fit.
### 3. Method Naming Issues
* **Confusing Introduction of Methods:** The paper introduces three methods (RL-SMC, SMCTS, TSMCTS). It does not make sense to name every iteration of the proposed methods. The paper should propose one method and investigate variations of it to understand the contributions of each change. This risks confusing the literature further, and it is unlikely that future work would use all of these ablations as baselines—just the final, aggregated method. Therefore, only this should be named.
### 4. SPO vs TRT SMC
* In multiple areas of the paper [4] is incorrectly referenced, such as in the experimental setup section. De Vries et al [4] is an iteration upon SPO which introduced the full experiment setup, environments and baselines, from which De Vries uses to investigate benefits to alternative SMC approaches. [4] should only be cited when referencing the specific SMC innovations used in their paper and SPO otherwise.
* SMC used as a policy improvement operator introduced in SPO needs to be made more clear in the related work section, this is a significant paper in the timeline of Sequential Monte Carlo for RL demonstrating not only policy improvement but the benefits of parallelism and competitiveness with MCTS.
---
### References
[1] Lazaric, A., Restelli, M., & Bonarini, A. (2007). Reinforcement learning in continuous action spaces through sequential monte carlo methods. In *Advances in Neural Information Processing Systems 20*.

[2] Macfarlane et al. SPO: Sequential Monte Carlo Policy Optimisation

[3] Piché et al. Probabilistic Planning With Sequential Monte Carlo Methods

[4] de Vries et al. Trust-Region Twisted Policy Improvement

**Questions:**

- What is the performance of the method from Piche et al (following SPO paper could be referred to as SMC-ENT), SPO and TRT-SMC for all experiments and how does it compare to the final method with all modifications, proposed in this paper

---

> ### Author Response · Authors · 2025-11-15
>
> Dear reviewer,
>
> We thank you for your time and many valuable comments.
> We address each of the concerns below.
>
> **Choice of baselines:**
>
> We thank the reviewer for pointing out that our presentation of the evaluation was not sufficiently clear.
> Please see the main rebuttal along with the changes in the document (Section 7, marked in green).
>
> **Coverage of Prior Work:**
>
> The focus of this section is to cover prior work on SMC *for search* in RL.
> Many prior works have applied SMC to decision problems, MDP solving, RL, state estimation in POMDPs, etc., and fall outside of the scope of this section.
> For example (beyond the work of Lazaric) see [2,3,4,5].
> We thank the reviewer for pointing out that this could have been made more clear, and have clarified this in the paper (Section 6, marked in green).
>
> **Claims of Parallelism:**
>
> The title was designed to emphasize the connection between *tree* search (exemplified in the *search, expansion, backpropagation* process of MCTS) and SMC and its parallel-particle based nature for search (*expansion, correction (/search)* and even *backpropagation* in T/SMCTS).
> If the reviewer finds the title as implying otherwise (which is valuable feedback), we will change it.
> We've edited it to "Twice Sequential Monte Carlo for Tree Search".
> We have also made the referencing to the demonstration of parallelism by SPO more explicit (Section 1, marked in green).
> If the concern remains unaddressed, please let us know in the discussion.
>
> **Motivation for Decoupling from CAI**:
>
> Thank you for pointing out that this point was not sufficiently elaborated.
> The motivation is as follows:
> The CAI-based derivation is less general in that it limits the search to one specific policy improvement operator.
> In contrast, the outside-of-CAI derivation accepts any policy improvement.
> The CAI derivation is far more complex: the seminal work of Piche [6] compared to the first paragraph of Section 3 in this work.
> Additionally, the derivation that is anchored in general policy improvement further expands the connection between SMC and policy improvement, demonstrating that it can be used for policy improvement without necessitating the non-trivial setup of Expectation Maximization.
> Finally, it facilitates the perspective shift from reasoning over a distribution over trajectories to reasoning over the values of actions from a mixture of improved policies at the root.
> We have elaborated the motivation at the opening of Section 3 (green) following this discussion.
>
> **Method names:**
>
> As the focus of this work is to improve on the planner, to facilitate clearer presentation we opted to give a name to each planner that is repeatdly referred to throughout the document.
> The operator-generalized, non-CAI *RL-SMC* is used to distinguish from Piche's original *CAI-SMC*.
> *SMCTS* is a subroutine used by the final algorithm. Since the subroutine is called by the final algorithm, it needs a name.
> Finally, *TSMCTS* is the final search algorithm.
>
> **SPO and TRT-SMC:**
>
> We thank the reviewer for pointing out that our referencing could have been made more accurate.
> We have edited the referencing to SPO and TRT [4] in Section 6 (Related Work, opening), 7 (Experiments, see main rebuttal) and 2 (Background, the end of the SMC paragraph).
>
> **Questions:**
>
> Please see the main rebuttal.
>
> Please let us know if any of the concerns remain unaddressed.
>
> References:
>
> [1] de Vries, et al., "Trust-Region Twisted Policy Improvement." ICML 2025.
>
> [2] Hoffman et al., "On solving general state-space sequential decision problems using inference algorithms". 2007.
>
> [3] Drovandi, et. al., "Sequential Monte Carlo for Bayesian sequentially designed experiments for discrete data." Computational Statistics \& Data Analysis, 2013.
>
> [4] Van der Vaart, et al. "Bayesian ensembles for exploration in deep q-learning." The Sixteenth Workshop on Adaptive and Learning Agents. 2024.
>
> [5] Abdulsamad et al. "Sequential Monte Carlo for Policy Optimization in Continuous POMDPs". Arxiv, 2025.
>
> [6] Piché, et al. "Probabilistic planning with sequential monte carlo methods." ICLR 2018.

---

> > ### Comment · Reviewer_oJ94 · 2025-11-19
> >
> > Thank you for the response and the changes made.
> >
> > Unfortunately, the rebuttal still does not directly address my fundamental concern about the missing standard baselines.
> > It remains completely unclear whether your final method (TSMCTS with all proposed modifications) actually outperforms the established prior SMC-based planners such as:
> >
> > - the SMC algorithm from Piché et al.
> > - SPO (Macfarlane et al.), and
> > - TRT-SMC (de Vries et al).
> >
> > It is perfectly acceptable (and useful) to include an ablation in which you replace only the SMC planning component while keeping everything else fixed.
> > However, this cannot replace a proper comparison against previous algorithms.
> >
> > At present, the key result tables and figures still lack these crucial baselines.
> > Without them, there is no evidence that TSMCTS advances the state of the art rather than just ablating components of your own method.

---

> > > ### Author Response · Authors · 2025-11-22
> > >
> > > We thank the reviewer for their reply.
> > >
> > > **TLDR: we emphasize that TSMCTS *is* compared to and outperforms both established prior SMC-based methods TRT-SMC (Figure 4) and SPO (Figures 1,2,3 and 5). Comparing TSMCTS to Piche's method is unnecessary to evaluate the contributions of this paper.**
> > >
> > > **Comparison to TRT-SMC [1]:**
> > >
> > > We emphasize that in Figure 4 included in the previous draft TSMCTS is compared to the full TRT-SMC [1] ("SMC + TRT" in the legend).
> > > We refer to this agent as "SMC + TRT" to emphasize that:
> > > (i) The approaches are modular from each other.
> > > (ii) The "+TRT" contributions are orthogonal and incorporable to those of TSMCTS (which is also shown in Figure 4).
> > > (iii) The only difference between this agent and TSMCTS is the planning procedure, which includes [1]'s contributions on top of the SMC backbone.
> > > If the source of the confusion is the name in the legend, we will change it from "SMC + TRT" to "TRT-SMC".
> > >
> > > **Comparison to SPO [2]:**
> > >
> > > We emphasize that the "SMC baseline" implemented by [1] to which we compare *is essentially* SPO.
> > > It is the latest variation to our knowledge of the model based agent which uses SMC for policy improvement through search (Algorithm 1 with SMC as the planner).
> > > The differences from the original SPO as proposed by [2] are minor:
> > > The "SMC baseline" uses more modern value targets (the root of the search vs. the prediction of the value network as a bootstrap), simplified outer loop (removed dynamic temperature etc.), simplified network architectures, etc..
> > > [1] refers to this agent as SMC in the legend to emphasize that the only difference between the agents in the plot is the planning procedure, a design choice which we follow.
> > > If the source of the confusion is the name in the legend, we will change it from "baseline SMC" to "SPO".
> > >
> > > **To alleviate any possible concerns that the implementation of [1] is not sufficiently comparable to the original SPO we include additional results with the official implementation of SPO from the *EdanToledo/Stoix* github repository in Figure 5 in the new draft.**
> > > We include results with the continuous environments and Snake, as a RubiksCube configuration does not appear to be implemented by Stoix.
> > > We include results with the original HPs for the continuous / discrete environments respectively.
> > >
> > > For the same compute resources, in all environments, the implementation of [1] is comparable or exceeds the performance of the original SPO. Respectively, **TSMCTS singificantly outpeforms or matches both in all environments**. This addresses the concern that the "SMC baseline" is not a sufficiently informative baseline to compare against.
> > >
> > > **Comparison to Piche's SMC [3]:**
> > >
> > > We note that:
> > >
> > > 1. The contributions of our paper are about *improving SMC as a search (planning) procedure for policy improvement in RL, by reducing variance and addressing path degeneracy at the root.*
> > >
> > > 2. [3] used SMC only for action selection, not for policy improvement, and thus is not a relevant baseline to start with.
> > >
> > > 3. The planners used by SPO and by [3] are essentially the same planner, where the only difference is a minor deviation in the objective (target) being estimated by the search. For that reason, by comparing to the *SMC planner* used by SPO we are indirectly comparing to the *SMC planner* used by Piche.
> > >
> > > 4. *[3]'s method was convincingly outperformed by SPO in the same environments in which TSMCTS is compared to SPO*. It is unnecessary to include every single historical RL method. Similarly, recent work improving on SMC as a search operator for policy improvement also omitted it [1].
> > >
> > > For these reasons, including [3] in the evaluation will not contribute to the conclusion of the paper: *TSMCTS represents a new state of the art SMC planner for policy improvement in RL*.
> > >
> > > We conclude that TSMCTS *is* compared to and outperforms both established prior SMC-based methods TRT-SMC (Figure 4) and SPO (Figures 1,2,3 and 5) and that comparing to Piche's method will not contribute to the existing evaluation.
> > >
> > > We thank the reviewer for the fruitful discussion.
> > >
> > > [1] de Vries, et al. "Trust-Region Twisted Policy Improvement." ICML 2025.
> > >
> > > [2] Macfarlane et al. "SPO: Sequential Monte Carlo Policy Optimisation." NuerIPS 2024.
> > >
> > > [3] Piché, et al. "Probabilistic planning with sequential monte carlo methods." ICLR 2018.

---

> > > > ### Comment · Reviewer_oJ94 · 2025-11-23
> > > >
> > > > Thank you for the detailed response.
> > > >
> > > > Regarding the SMC baseline: if it is essentially SPO (with different network architectures and value approximations), I believe it should be clearly labeled as such. I think introducing an arbitrary SMC baseline is unhelpful for tracking progress in this field. I do not consider the network architecture or value approximation to be a particularly distinctive or important part of this baseline (of course minimising discrepancy and using the same value approximations would be preferable), so this baseline should simply be referred to as SPO, in a similar way to how PPO has been maintained as a baseline through the literature.
> > > >
> > > > The key modification however appears to be the removal of dynamic temperature scheduling and the use of a fixed temperature instead. I do believe this is a substantial change. In my view, this baseline should therefore be described as “SPO w/ fixed temperature.” Moreover, dynamic temperature scheduling may in fact be an important component of SPO’s performance, especially since the original SPO paper conducts a hyperparameter search over temperature and shows that performance is highly sensitive to this choice. I note that this paper requires a hyperparameter search over temperatures to find a suitable value, something that this adaptive temperature avoids. I strongly encourage the authors to include in addition, the full SPO baseline with dynamic temperature in the comparison in Figure 1.
> > > >
> > > > After reflecting on your explanation, I agree that a Piche et al. baseline is not required, as it falls outside the scope of algorithms that use SMC for policy improvement.
> > > >
> > > > On the experimental setup in Figure 1: I have also noticed that the first two environments are evaluated for 1e7 environment steps, whereas Ant, Humanoid, and HalfCheetah are trained for only 1e6 steps. Could the authors explain this discrepancy? In particular, I note that the baselines do not appear to have converged at 1e6 steps, which raises the question of why these experiments were not all run for the same number of steps, 1e7.

---

> > > > > ### Author Response · Authors · 2025-11-23
> > > > >
> > > > > We thank the reviewer for acknowledging our rebuttal.
> > > > >
> > > > > **Number of timesteps in the DM control environments:**
> > > > >
> > > > > *TLDR: The experiments were ran for the popular number of timesteps in these environments in literature.*
> > > > >
> > > > > First, please note that the experiments were ran until 5e6, not 1e6.
> > > > > The popular number of steps for experiments in Mujoco / continuous DM control is usually between 1e6 and 5e6. For examples see recent works such as [1], [2] and TD7 [3] as well as seminal works such as TD3 [4] and SAC [5]. In this case, for simplicity and uniformity we followed exactly the choices made by the recent prior work of TRT SMC [2].
> > > > >
> > > > > **Presentation:**
> > > > >
> > > > > We thank the reviewer for the useful suggestions.
> > > > > We have reworked the presentation in the experimental setup to be in line with the comments of the reviewer:
> > > > >
> > > > > 1. Figure 1 now presents TSMCTS (and TSMCTS + TRT) in comparison to existing popular baselines (SPO, TRT SMC, PPO, etc.), which demonstrates the answer to the first question: TSMCTS represents a new SOTA in this area.
> > > > >
> > > > > 2. Figure 2 now explicitly ablates a comparison between the three planners, using the same implementation for the outer loop: the baseline SMC, TSMCTS, and GumbelMCTS. TRT SMC is omitted as Figure 1 shows that its contributions are orthogonal to the TSMCTS backbone.
> > > > >
> > > > > 3. Figure 3 follows in the same manner, vs. runtime.
> > > > >
> > > > > 4. Figure 4 demonstrates that the claims of: (i) variance reduction, (ii) address of path degeneracy at the root and (iii) scaling with sequential compute are all supported empirically.
> > > > >
> > > > >
> > > > > With this, we believe we have addressed all concerns raised by the reviewer in the review and throughout the rebuttal. If this is not the case or if there are other reservations we are looking forward to address them.
> > > > >
> > > > > [1] Oren et al., Value Improved Actor Critic Algorithms. NeurIPS 2025.
> > > > >
> > > > > [2] de Vries, et al. "Trust-Region Twisted Policy Improvement." ICML 2025.
> > > > >
> > > > > [3] Fujimoto et al., For SALE: State-Action Representation Learning for Deep Reinforcement Learning. NeurIPS 2023.
> > > > >
> > > > > [4] Fujimoto et al., Addressing Function Approximation Error in Actor-Critic Methods. ICML 2018.
> > > > >
> > > > > [5] Haarnoja et al., Soft actor-critic: Off-policy maximum entropy deep reinforcement learning with a stochastic actor. ICML 2018.

---

> > > > > > ### Comment · Reviewer_oJ94 · 2025-11-25
> > > > > >
> > > > > > I thank the authors for there continued improvements of the paper following discussions.
> > > > > >
> > > > > > Please see the above comments regarding the need to specify SPO with a fixed temperature. I still maintain my recommendation to include the full baseline in the main results, using the dynamic temperature as an additional baseline.
> > > > > >
> > > > > > Thank you for the comments explaining the decision to use 5e6 transitions for these benchmarks. I think it is reasonable to evaluate at this level of compute. However, the paper must clearly acknowledge that these results use a far lower transition count compared to [1], which uses 1e8 transitions and trains to convergence.
> > > > > >
> > > > > > Since the present work neither trains to convergence nor matches the compute level of [1], I do not believe it is appropriate to claim that the proposed method fully outperforms SPO. The conclusions should therefore be toned down to recognise this limitation and explicitly state that future work should investigate higher-compute regimes to determine whether the observed gains persist at such scales.
> > > > > >
> > > > > > [1] Macfarlane et al. “SPO: Sequential Monte Carlo Policy Optimisation.” NeurIPS 2024.

---

> ### Author Response · Authors · 2025-11-26
>
> We thank the reviewer for the additional comments and for clarifying the remaining concerns.
>
> **Labeling SPO with fixed temp.:**
>
> We are hesitant to label the "SMC baseline" as "SPO w. fixed temp." for the following reasons:
>
> 1. There are differences between [1]'s simplified implementation of SPO (the "SMC baseline") and the original SPO beyond the dynamic temperature:
> (i) The original SPO used trust regions when training the policy (i.e. in the policy loss), which are omitted by [1].
> (ii) The original SPO used Dirichlet noise which is omitted by [1].
> For these reasons, labeling the SMC baseline as SPO w. fixed temp. could be considered insufficiently accurate.
>
> 2. The purpose of Figures 2-3-4 is to *explicitly ablate the planning procedure* (SMC, TSMCTS, GumbelMCTS) rather than the complete algorithm (SPO, AZ).
> All agents in these figures are *identical*: losses, targets, training procedure, etc. The only difference between them is the planner.
> For this reason we also label the MCTS agent as GumbelMCTS rather than GumbelAZ in these figures.
> The purpose of this comparison is to show that all else being equal, TSMCTS is the best performing *planner* for policy improvement in these environments.
> This is clarified in *lines 432-441*.
> By labeling the "SMC baseline" as "SPO w. fixed temp.", it is less clear that the comparison is between planners, not between methods.
>
> 3. From our perspective, it would be as correct to think of TSMCTS as an SPO variant, as it would be to think of AZ with more advanced MCTS variations as still AZ. Respectively, one could think of "TSMCTS" as "AZ with TSMCTS instead of MCTS".
> Labeling the "SMC baseline" as "SPO w. fixed temp." and the "TSMCTS" agent as "TSMCTS" suggests that this is not the case, which we think undersells the contributions of SPO: that SMC can be used for policy improvement.
>
> We clarify that the "SMC baseline" is essentially a simplified SPO agent, and strengthen the connection to AZ, in *lines 436-440*.
> Please let us know if this addresses this point, or if despite the above motivation the reviewer would like the "SMC baseline" to explicitly be labeled as "simplified SPO" or "SPO w. fixed temp.".
>
> **Including the "full baseline" in the main results:**
>
> We assume that by the "full baseline" the reviewer refers to what is currently labeled as the "SMC baseline"? We note that the *full-SPO baseline (Stoix)* is included in the main results, Figure 1.
>
> Currently, Figure 1 contains familiar baselines (i.e. PPO,GumbelAZ, SPO, TRT SMC) and Figures 2-4 contain planner ablations.
> Given the previous point about "SPO with fixed temp.", please let us know if the reviewer finds it important that the "SMC baseline /  simplified SPO" curve is included additionally in Figure 1 (we note  the comparison between these agents is currently presented in Figure 2), or if the reviewer is satisfied with the current structure.
>
> **Clarifying transition count:**
>
> We thank the reviewer for the comment.
> We've added, *line 407* in the latest draft uploaded with this reply.
>
> **More conservative conclusions:**
>
> We are in agreement with the reviewer: In the paper the conclusions are more conservative.
>
> The conclusion from Figure 1 (line 421-422) is currently:
> *In all environments the TSMCTS-based agents outperform or match all baselines.*
>
> Respectively, the conclusion from Figure 2 (line 441 442) is:
> *TSMCTS is overall the dominant search operator for policy improvement compared to both the SMC baseline and GumbelMCTS in these experiments.*
>
>
>
> We thank the reviewer again for the fruitful discussion. Please let us know if anything remains unaddressed.
>
> [1] de Vries, et al. "Trust-Region Twisted Policy Improvement." ICML 2025.

---

> > ### Comment · Reviewer_oJ94 · 2025-11-27
> >
> > I thank the reviewers for their response.
> >
> > Just to clarify, when I referred to a “full baseline” in my previous comment, I meant adding a baseline of SPO that uses dynamic temperature (as an additional baseline) to the current figures, in addition to the existing baseline now called SPO, which was previously referred to as SMC-baseline. I do not think this is a strict requirement however for acceptance.
> >
> > Thank you for explaining the full differences between the SPO baseline you have implemented and the one with dynamic temperature + trust region loss. I am satisfied that this is sufficient for comparison, as the differences are now clearly stated. I also agree that the naming can be left as it is given these differences. I also note that the authors have added a line discussing transition counts, which addresses my concerns regarding fair comparison to other works.
> >
> >
> > Given all the changes made by the authors that go a significant way to addressing my main concerns I will upgrade my score to a 6. I believe that my issues with the soundness of the paper have been largely addressed and will update this to a 3.

---

> ### Author Response · Authors · 2025-11-28
>
> We thank the reviewer for acknowledging our rebuttal, the openminded discussion and  the many valuable comments.
>
> We clarify that **the SPO baseline In Figure 1 is *not* the SMC baseline:  it is the full SPO agent from the official implementation of Stoix**, using the HPs from the SPO paper - with dynamic temperature, trust regions, etc., as suggested by the reviewer.
>
> The “SMC baseline” is now exclusively included in the planner-ablations, Figures 2-4.
>
> This is intended to fully address the reviewer’s point - with which we agree - that it is important to compare to the original SPO agent, as well to explicitly ablate the planners. If this does not address it, please let us know.

---

### Official Review · Reviewer_WwXY · 2025-11-01

**Soundness:** 2
**Presentation:** 2
**Contribution:** 2
**Rating:** 4
**Confidence:** 3

**Summary:**

The paper proposes Twice Sequential Monte Carlo Tree Search (TSMCTS), a planner built on a reformulation of SMC for RL (RL-SMC/SMCTS) that explicitly targets policy improvement at the root rather than trajectory inference. The authors prove that RL-SMC becomes a policy-improvement operator in the infinite-particle limit, then address two classical SMC issues—exploding variance with depth and path degeneracy—by integrating Sequential Halving (SH) at the root and aggregating SMCTS value estimates across SH rounds. Empirically, across discrete and continuous benchmarks, TSMCTS outperforms an SMC baseline and GumbelMCTS, with lower estimator variance, mitigated target degeneracy at the root, and better scaling with sequential compute while retaining SMC’s parallelization benefits.

**Strengths:**

- Clear conceptual shift. The paper reframes SMC for root-focused policy improvement, pairing importance-weighting/backprop with a value-based “SMCTS” that more closely mirrors MCTS, then layers SH to reduce variance and degeneracy at the root. The pipeline is well-motivated and algorithmically coherent.

- Parallelization narrative. The introduction positions SMC as easier to parallelize (GPU-friendly) than MCTS due to MCTS’s sequential nature and tree-memory overhead—useful context for why this line is promising.

- Addresses core SMC pain points. The paper explicitly diagnoses variance growth with depth and path degeneracy, then justifies why SH (known budget, repeated resets, per-action parallelism) should help at the root.

**Weaknesses:**

- Finite-sample guarantees are thin. The key theory (policy improvement) is stated for infinite particles; practical behavior with finite
N, finite evaluation accuracy, and shallow search is not quantified with explicit bias/variance or MSE bounds as functions of depth and budget.

- Comparative scope. Beyond GumbelMCTS, stronger or modernized MCTS baselines (e.g., robust PUCT variants, MuZero-style planners under equalized compute/memory) are not deeply explored; reproducibility notes exist, but compute normalization remains largely wall-clock-based, which can be hardware-dependent.

- Root-centric metrics. Degeneracy is principally assessed at the root (active actions). It is less clear how diversity behaves down the tree in deeper horizons, particularly for large or continuous action spaces.

- Trade-off analysis. While SH intuitively reduces variance, the paper does not provide formal rates showing how SH-aggregation plus per-action budget reallocation improve the estimator of the root value or decision quality relative to SMCTS/SMC at fixed compute.

**Questions:**

1. Finite N theory: Can you provide finite-particle error bounds (bias/variance or MSE) for the root value estimator and/or the action selection error under TSMCTS, explicitly showing dependence on depth and per-round budgets, and contrasting with SMCTS/SMC?

2. Robustness to poor priors. Early SH rounds may prune good actions if the policy prior is miscalibrated. Do you employ safeguards (temperature/bonuses or randomized inclusions) to prevent premature elimination? How sensitive is TSMCTS to prior entropy?

3. Beyond the root. Do you track effective sample size/diversity at deeper nodes to confirm that degeneracy is not merely shifted downward? If so, please include these diagnostics; if not, what is your expectation theoretically?

4. SH-induced bias. Since SH repeatedly resets to the root and aggregates across rounds, can you characterize the bias–variance trade-off more formally (e.g., a bound on regret at the root vs. horizon depth), and conditions under which SH yields decision-consistent improvements?

---

> ### Author Response · Authors · 2025-11-15
>
> Dear reviewer,
>
> Thank for your detailed review and the many comments and questions.
>
> **1. Finite particle / depth guarantees:**
>
> The exact variance will be dependent on the learning dynamics of the neural network, prior policy and MDP dynamics and remains an open problem in RL in general.
> For these reasons, the seminal work by Piche [1] does not provide any theoretical guarantees or analysis (finite or in expectation).
> SPO [2] which was the first to use SMC for search for policy improvement provides guarantees only in expectation, similar to our Thm 1.
> The more recent work by [3] does not provide any theoretical motivation or guarantees.
> GumbelMCTS [4] proves only that $I_{GMZ}$ provides policy improvement.
>
> In comparison, in addition to theoretical guarantees in in expectation (Thm 1), we provide results on the effects of finite depth $T$ (Corollary 1), as well as an extensive theoretical discussion over variance reduction in Appendix A.3.
>
> We can however add the following:
>
> *Finite depth*: From Thm and Corollary 1, we know that in expectation, the improvement increases with $T$.
> The increase is bounded by the maximum local improvement possible at each step. We note that the variance increases with $T$ as well (see Section 3).
>
> *Fixed depth, finite particles*: an established result for SMC algorithms of this family is that the variance of the estimator reduces with $1/N$, where $N$ is the number of particles ([5], Theorem 1).
>
> *Uncertain value function*: even for well established, well investigated MCTS, only very recently an approach that theoretically analyzes and accounts explicitly for the variance of a learned value function was proposed [6]. We leave a similar analysis in SMC methods for future work.
>
> **2. Comparative scope**:
>
> Our aim is to compare to similar approaches for policy improvement through search.
> One of the strengths of our method is that advanced search policies such as robust PUCT and others can be directly incorporated - the only requirement is that they be policy improvement operators.
> GumbelMCTS is a MZ style planner (the latest along this line from the same authors, to our knowledge), which is the reason for its inclusion.
>
> **3. Root-centric metrics**:
>
> The focus of this work is to address the effects of path degeneracy *explicitly* at the root to achieve better policy improvement at the root. As this is the focus, the developed mechanisms are only investigated with respect to these effects (i.e. at the root).
> We note that path degeneracy at the children of the root is essentially guarenteed, and not immediately detrimental: it induces focus on promising larger-value trajectories for the value prediction $Q(s_{root},a)$.
>
> **4. Trade-off analysis**:
>
> Neither does the seminal work [4] to our knowledge representing the current SOTA for AZ/MZ style planners for policy improvement and the first to incorporate SH at the root for search.
> [4] motivates SH as a simple-regret-minimization (pure-exploration) algorithm.
> While this argumentation applies to our work (even more strongly, because in TSMCTS the "arms" are stationary, as assumed by SH, which they are not in MCTS), we add the variance-reduction effects to the motivation (see Appendix A.4 for more detail).
> We also include a concrete empirical investigation of the variance-reduction effects of SH (Figure 3, center) to verify the effect empirically.
>
> **Questions:**
>
> 1. See 1.
>
> 2. SH provides theoretical guarentees to address this concern, see [7].
> For that reason, TSMCTS is the *least* sensitive to prior entropy, between SMC, SMC+TRT and T/SMCTS through larger num. of searched actions resulting in signif. less variance (Figure 3 center) compared to the other variants.
> It is possible to employ any additional regularization to the prior (such as entropy), orthogonally to our contributions.
> Figure 3 right suggest that TSMCTS makes this less necessary than for previous variations.
>
> 3. See 3.
>
> 4. SH does not induce bias. The SMCTS-based search of actions at the root independently in parallel induces a true stationary bandit with unbiased predictions for the arms, for which SH comes with "almost-optimal"ity theoretical guarentees, see [7]. Does this answer the reviewer's question?
>
> Please let us know if any questions or concerns remain unaddressed.
>
> References:
>
> [1] Piché, et al. "Probabilistic planning with sequential monte carlo methods." ICLR 2018.
>
> [2] Macfarlane, et al. "SPO: Sequential Monte Carlo Policy Optimisation." NuerIPS 2024.
>
> [3] de Vries, et al. "Trust-Region Twisted Policy Improvement." ICML 2025.
>
> [4] Danihelka, et al. "Policy improvement by planning with Gumbel." ICLR 2022.
>
> [5] Chopin. "Central limit theorem for sequential Monte Carlo methods and its application to Bayesian inference." The Annals of Statistics, 2004.
>
> [6] Oren, et al. "Epistemic Monte Carlo Tree Search." ICLR 2025.
>
> [7] Karnin, et al. "Almost optimal exploration in multi-armed bandits." ICML, 2013.

---

### Official Review · Reviewer_t5vm · 2025-11-01

**Soundness:** 3
**Presentation:** 1
**Contribution:** 3
**Rating:** 4
**Confidence:** 3

**Summary:**

This paper proposes an advanced planning method, SMCTS, which integrates the backpropagation mechanism of MCTS into the classical SMC framework. To further enhance the root policy, the authors introduce TSMCTS, which employs Sequential Halving to focus search resources on a progressively smaller set of actions and runs SMCTS in parallel for each action. TSMCTS reduces variance through Q-value averaging and the SH mechanism while preserving the inherent parallelization advantages of SMC methods.

**Strengths:**

1.The authors present their method systematically, demonstrating step by step how their algorithm design addresses the limitations of existing approaches.
2. Combining the backpropagation mechanism of MCTS with the SMC framework is a novel and interesting idea. The inherent parallelization of SMC also shows strong potential for accelerating planning algorithms.
3. Unlike traditional MCTS, TSMCTS naturally supports both discrete and continuous environments, which is a significant step toward developing a more universal planner.

**Weaknesses:**

1. The paper includes extensive background material and relies heavily on prior works to explain the proposed methods. This makes it difficult to fully understand certain design choices without consulting the references. I suggest presenting a more self-contained description of the algorithms and moving some background discussions to the related work section.
2. There are several typographical errors and inconsistencies in the notations, suggesting that the notation system could benefit from more careful refinement.
3. The experimental section lacks detailed setup descriptions and thorough result analysis, particularly for the main experiments. The authors should explain the rationale behind the choice of environments, discuss why performance differs across them, and consider evaluating on more than five environments to strengthen the empirical support.

**Questions:**

1. Why does TSMCTS fail to outperform GumbelMCTS on the Rubik’s Cube task? Is this due to characteristics of the environment, or does TSMCTS generally lose its advantage on more complex tasks?
2. Please discuss the potential of applying TSMCTS to visual-observation tasks such as Atari, or to problems without an explicit model?
3. If my understanding of section 2 is correct, a policy improvement operator typically acts on a policy together with its value function. However, in Equation (16), the current policy is paired with the value of the improved policy. Could the authors please explain the theoretical justification for this formulation?
4. Please discuss how to choose the parameter $\beta$ in Equation (4)?
5. In my understanding, GumbelMCTS is designed for discrete tasks originally. How do the authors apply it to continuous tasks?
6. Other  issues:
	In Equation (4), $\pi$ and $\pi_\theta$ seem to represent the same quantity. Similar inconsistencies appear elsewhere.
	In line 124, the state is equated to a probability, which seems incorrect or unclear.
	In line 128, what does $k$ represent? Likewise, in line 143, the meaning of $Q_i$ is unclear.
	in equation 7 and 8, the $A_soft$ is given in different forms.
	Algorithm 1 is never explicitly referenced in the text.
	In algorithm 2, I can't find where ancestor identifier is defined. And if the loop starts from $t=1$, $s_1$ is undefined. The same issue appears in Algorithm 3.
Overall, I suggest the authors carefully review the entire paper to correct unclear or inconsistent notations and definitions. At the moment this is one of the key points resulting in the selected score.

---

> ### Author Response · Authors · 2025-11-15
>
> Dear reviewer,
>
> Thank you for your valuable time, detailed review, valuable comments and questions.
>
> **Experimental section:**
>
> We thank the reviewer for pointing this out. Please see the main rebuttal.
>
> **The paper includes extensive background material and relies heavily on prior works to explain the proposed methods. This makes it difficult to fully understand certain design choices without consulting the references.**
>
> Could the reviewer specify the design choices that are not explained in a sufficiently well contained manner?
>
> **There are several typographical errors and inconsistencies in the notations, suggesting that the notation system could benefit from more careful refinement.**
>
> We thank the reviewer for the valuable comments.
> We agree with the observations listed by the reviewer in question 6. We upload an editted version with the rebuttal, with changes marked in green.
> Please let us know if anything remains unclear / any questions or concerns remain in that regard.
>
> **Questions:**
>
> 1. Ultimately, it is unclear. This is the reason prior work - and us - include multiple environments in the evaluation, to observe the behavior across different domains.
> It is worth mentioning that MCTS spends more compute per expansion (see Appendix A.5), and since the resources are equated per-expansion in Figure 1, MCTS should be expected to have the potential for better performance per-expansion.
> We note that the main objective of this work is to demonstrate improvement over the SMC baseline, which it improves on in every domain.
>
> 2. [1] applied SMC successfully on a learned model.
> As TSMCTS iterates on the search used by this prior work, we would expect it to work just as well.
> Similarly, we note that MCTS was very successful across board-game observations, known model (AlphaZero, [2]) as well as Atari, visual observations, latent space learned model (MuZero [3]).
> Due to the similarities between AlphaZero and SPO [4] (which we frame as an RL learning loop with modular MCTS / SMC search, Algorithm 1), we would expect the SMC variations - including TSMCTS - to work well for policy improvement also in latent-space models.
> This hasn't yet been shown empirically by prior work to our knowledge however and goes beyond the scope of this work.
>
> 3. We agree with the reviewer's observation.
> First, we note that the original work which introduced this approach for policy improvement at the root of the search tree, GumbelMCTS [6], did so without theoretical motivation in this regard.
> That being said, recent work makes it possible to provide theoretical motivation for this: [7] prove that as long as the values satisfy $Q^{\pi'}(s,a) \geq Q^{\pi}(s,a)$ (i.e. the value with respect to which we do improvement, $Q^{\pi'}$, is itself of an improved policy), it is sound to improve $\pi$ with respect to $Q^{\pi'}$ (Theorem 3 and Corollary 2 in [7]).
> This is of course the case in T/SMCTS.
> We will add this discussion to the paper.
>
> 4. These parameters should be tuned with HP tuning. It is essentially the same parameter used by GumbelMCTS as well as [1, 4, 5], and is tuned by these prior works in the same manner. See Appendix D, Hyperparameters paragraph, for details on how it was tuned for T/SMCTS.
>
> 5. We use the same baseline as [5], which uses the approach of SampledMuZero [8] to apply MCTS to continuous spaces. The core idea is to sample N actions from the prior policy, and treat the action space as discrete in these actions (this was originally described briefly in the environments paragraph in Appendix D, and is now extended in Appendix B to make it more clear).
>
> 6. We thank the reviewer again for the valuable comments.
>
> Please let us know if any concerns or questions remain unaddressed.
>
> References:
>
> [1] Piché, et al. "Probabilistic planning with sequential monte carlo methods." ICLR 2018.
>
> [2] Silver et al. "Mastering Chess and Shogi by Self-Play with a General Reinforcement Learning Algorithm." Science, 2018.
>
> [3] Schrittwieser et al. "Mastering atari, go, chess and shogi by planning with a learned model." Nature 2020.
>
> [4] Macfarlane, et al. "SPO: Sequential Monte Carlo Policy Optimisation." NeurIPS 2024.
>
> [5] de Vries, et al. "Trust-Region Twisted Policy Improvement." ICML 2025.
>
> [6] Danihelka, et al. "Policy improvement by planning with Gumbel." ICLR 2022.
>
> [7] Oren, et al. "Value Improved Actor Critic Algorithms." NeurIPS 2025.
>
> [8] Hubert, et al. "Learning and planning in complex action spaces." ICML, 2021.

---

### Author Response · Authors · 2025-11-15

We thank the reviewers for their time, detailed reviews and many valuable comments.
We include a dedicated rebuttal for each reviewer, answering their questions and specific concerns, but want to address a few points that were shared among reviewers first.

**Presentation of the evaluation:**

It seems that the presentation of the evaluation in the original draft were not sufficiently clear. We thank the reviewers for pointing that out and include an edit with clarified evaluation (see Section 7). Changes are marked in green to make them easy to identify. We include a brief summary of the evaluation and its motivation below:

*Motivation for the experimental setup:*

The objective of our work (TSMCTS) is to improve the SMC planner used by SPO [1] and TRT-SMC [2] for policy improvement in RL, by addressing the problems of *high variance* and *path degeneracy at the root* that SMC suffers from.
SPO established an experiment bed, with discrete and continuous action environments, where it demonstrates the strength of SMC for policy improvement.
TRT-SMC which is to our knowledge the latest SOTA SMC-based planner for policy improvement later iterated on this experiment bed.
We use *the same setup (environments and experiments, implementation and hyperparameters) used by TRT-SMC*, in order to keep the evaluation as scientific as possible.

*Evaluation:*

Our objectives are to :

1. Establish that TSMCTS indeed improves upon the latest SMC framework used by prior works ([1,2]) for policy improvement (the "SMC baseline") and compares or even outperforms a popular modern version of MCTS (GumbelMCTS [3], which is to our knowledge the latest along the AlphaZero [AZ, 6] / MuZero [MZ, 7] line).

2. Establish that TSMCTS is able to *scale* with increased sequential compute (depth), where the baseline *degrades* with increased depth.

3. Verify empirically that TSMCTS indeed reduces variance and addresses the problems of path degeneracy at the root in practice, in addition to theoretically.

Following discussion with reviewer oJ94, we add:

4. Verify that TSMCTS offers improvement over complete existing baselines, outside of the equalized-conditions domain.

For this purpose, we present 4 figures:

In Figure 1 we compare: (i) *New:* the **original SPO** from the official implementation [5]. (ii) *New:* **TRT-SMC**, (iii) **GumbelAZ** and (iv) model free **PPO** [4] for reference. We also include *new* **TSMCTS + TRT** to demonstrate that the TRT contributions of [2] are orthogonal to TSMCTS's and can be combined (as discussed in Section 6). **In all environments TSMCTS-based agents outperform or match all other baselines (objective 4)**.
We note that the twisting parameters of TRT were not re-tuned for TSMCTS, suggesting that it may be possible to increase the performance for the TSMCTS + TRT variant even further.

In Figure 2-3 we *explicitly ablate the planners* under equalized conditions. We do this by comparing agents that *only* differ in the planning procedure used for policy improvement: (i) **TSMCTS**, (ii) the **SMC baseline** (which can be thought of as *simplified-SPO*, or *TRT-SMC without the TRT contributions* or even as *AZ with SMC instead of MCTS*) and (iii) **GumbelMCTS** (which is the same GumbelAZ agent as in Figure 1, labelled this way to emphasize that the *only* difference between the agents is the planner). TRT-SMC is omitted as its contributions are shown to be orthogonal to the contributions TSMCTS in Figure 1. **TSMCTS is the dominant search operator for policy improvement (objective 1)**.

In Figure 4 we include additionally (iv) the intermediate **SMCTS** used by TSMCTS, to show that the contributions of Section 5 are significant, and
(v) the component of **TRT-SMC**'s contribution which aims to address path degeneracy at the root. GumbelMCTS, which is not relevant for this comparison, is omitted. **TSMCTS *scales* with depth where the baselines *degrade* (objective 2). TSMCTS significantly reduces variance compared to the baselines and better addresses the effects of path degeneracy at the root (objective 3).**

We thank the reviewers again for their valuable comments.

References:

[1] Macfarlane et al. "SPO: Sequential Monte Carlo Policy Optimisation." NuerIPS 2024.

[2] de Vries et al. "Trust-Region Twisted Policy Improvement." ICML 2025.

[3] Danihelka et al. "Policy improvement by planning with Gumbel." ICLR 2022.

[4] Schulman et al. "Proximal policy optimization algorithms." 2017.

[5] Toledo, Stoix: Distributed Single-Agent Reinforcement Learning End-to-End in JAX. Github, 2024.

[6] Silver et al. "Mastering Chess and Shogi by Self-Play with a General Reinforcement Learning Algorithm." Science, 2018.

[7] Schrittwieser et al. "Mastering atari, go, chess and shogi by planning with a learned model." Nature 2020.

---

### Meta-Review · Area_Chair_hrYc · 2026-01-06

**Summary:**

The manuscript considers a new variant of the sequential Monte Carlo method as an alternative for Monte Carlo Tree Search for reinforcement learning. Overall, while the proposed method seems interesting, the presentation and the comparison seem lacking. While the authors addressed partially those concerns, the meta-reviewer feels that the paper needs more polishing before it can be considered for acceptance.

**Reviewer Concerns:**

The authors partially addressed the concerns raised by the reviewers, while upon reading the manuscript, the meta-reviewer still finds the presentation need improvement.

**Reviewer Scores:**

I believe the reviewers will keep their score based on the interaction.

---

### Decision · Program_Chairs · 2026-01-26

Reject